

# Fourteen new species of *Oecetis* McLachlan, 1877 (Trichoptera: Leptoceridae) from the Neotropical region

Fabio B. Quinteiro[1] and Ralph W. Holzenthal[2]

[1] Departamento de Biologia, Faculdade de Filosofia, Ciências e Letras de Ribeirão Preto, Universidade de São Paulo, Ribeirão Preto, São Paulo, Brazil
[2] Department of Entomology, University of Minnesota - Twin Cities Campus, Saint Paul, MN, USA

## ABSTRACT

**Background:** The caddisfly genus *Oecetis* currently contains 534 valid species. Its larval stages are found in freshwaters around the world. The adults can be distinguished from other Leptoceridae by the unbranched forewing M vein and the exceptionally long maxillary palps. In the Neotropical region, 55 species of *Oecetis* have been recorded and most of them can be placed in one of the six species groups known from this biogeographical region: the *avara*-, *falicia*-, *inconspicua*-, *punctata*-, *punctipennis*-, and *testacea*-groups. More than 50% of the known diversity of Neotropical *Oecetis* has been described in the past 40 years. Here, we describe an additional 14 new species of *Oecetis* to further document the diversity of this genus in the Neotropical region.

**Methods:** The descriptions and illustrations presented here are based on male specimens. Specimens were collected with Malaise traps or ultraviolet light traps. They were preserved in alcohol or pinned as stated in material examined section. Specimens had their genitalia prepared in 85% lactic acid to better observe internal characters and illustrations were aided by the use of a microscope with drawing tube attached.

**Results and Discussion:** This study raises the number of species of *Oecetis* in the Neotropics from 55 to 69. Eight of the new species presented here could not be reliably placed in one of the known species groups (*Oecetis acuticlasper* n. sp., *Oecetis flinti* n. sp., *Oecetis carinata* n. sp., *Oecetis cassicoleata* n. sp., *Oecetis blahniki* n. sp., *Oecetis gibbosa* n. sp., *Oecetis licina* n. sp., and *Oecetis pertica* n. sp.). The others are placed in the *punctata*-group (*Oecetis bidigitata* n. sp., *Oecetis quasipunctata* n. sp.), *testacea*-group (*Oecetis plenuspinosa* n. sp.), and *falicia*-group (*Oecetis calori* n. sp., *Oecetis hastapulla* n. sp., *Oecetis machaera* n. sp.). Most of the diagnostic characters rely on structures of the inferior appendages and phallic apparatus, and the shape of tergum X.

Corresponding author
Fabio B. Quinteiro,
fabiobquinteiro@gmail.com

## INTRODUCTION

*Oecetis McLachlan, 1877*, is a genus within the caddisfly family Leptoceridae (Insecta: Trichoptera) with 534 valid species. The species in the genus are usually collected along

rivers and lakes, where they tend to be very abundant (*Schmid, 1998*). The genitalia are characterized by having segment IX usually short, the preanal appendages somewhat prominent and ovoid, and the inferior appendages single-segmented (*Schmid, 1998*). The adults can be distinguished from other caddisflies by their unique wing venation, with an unbranched forewing M vein (*Quinteiro & Calor, 2015*), as well as their exceptionally long maxillary palps (*Schmid, 1998*). Their color ranges from pale yellow to dark brown and their size is from 6 to 12 mm (*Henriques-Oliveira, Dumas & Nessimian, 2014*). The forewing often has characteristic brown spots.

YE Chen (1993, unpublished data) provided a comprehensive treatment of *Oecetis* in his unpublished PhD dissertation. However, the genus has never undergone a thorough revision since Chen's work was never published. Nevertheless, many authors produced works on more restricted groups of species from around the world (*Malicky, 2005*; *Rueda-Martín, Gibon & Molina, 2011*; *Lee, Hwang & Bae, 2012*; *Blahnik & Holzenthal, 2014*).

In the Neotropical region, specimens of *Oecetis* can be found in almost every freshwater environment, especially those with sandy substrate and slow running water. Fifty-five species of *Oecetis* have been recorded in the Neotropics (*Holzenthal & Calor, 2017*) and most of them can be placed in one of six species group (*Neboiss, 1989*; *Malicky, 2005*): *avara-*, *punctata-*, *falicia-*, *inconspicua-*, *punctipennis-*, and *testacea*-group.

The *avara*-group has the shape and structure of the inferior appendages somewhat mitten-like as its most diagnostic aspect (*Blahnik & Holzenthal, 2014*). Species from the *avara*-group are recorded from Canada to the north of South America (*Blahnik & Holzenthal, 2014*).

The *punctata*-group species are restricted to the Neotropical region (*Blahnik & Holzenthal, 2014*) and have the inferior appendages somewhat quadrate, often with apical processes bearing strong, thick setae.

The *falicia*-group is also endemic to the Neotropics and can be diagnosed by the dorsolateral processes on segment IX and the membranous tergum IX of the male (YE Chen, 1993, unpublished data).

The *inconspicua*-group is diagnosed by its phallic apparatus rounded and wide, with a pinched ventral projection and a helical phallic spine. This group was not properly addressed by YE Chen (1993, unpublished data); even so it is often referred informally in the literature. Its species were treated as two groups by Chen: *ochracea-* and *furva*-group. However, they both share the very similar phallic apparatus shape and even an inferior appendage with rounded dorsal lobe and acuminate distal lobe apex. Representatives of this group can be found in all biogeographic areas except the Australasian and Oriental (YE Chen, 1993, unpublished data). In the Neotropics, the species recorded in this group are *O. inconspicua* (Walker), *O. excisa* Ulmer, *O. pseudoinconspicua* Bueno-Soria, *O. amazonica* (Banks) and *O. pseudoamazonica*, Rueda-Martin, Gibon, Molina.

The *punctipennis*-group can be diagnosed by the forewing $R_{1+2}$ vein divided very close to the tip of the wing (YE Chen, 1993, unpublished data). This group seems to have a trans-Antarctic distribution (South America, Australia) but the species in the Neotropical region have, as an additional diagnostic character, the phallic apparatus, short, bent ventrally, and with a horseshoe-shaped phallotremal sclerite. In the Neotropics, the

included species in this group are: *O. punctipennis* (Ulmer), *O. iguazu* Flint and *O. connata* Flint.

The *testacea*-group is diagnosed as having a honeycomb microstructure covering abdominal tergum VIII and usually preceding terga, although their function is yet unknown (*Malicky, 2005*). *Henriques-Oliveira, Dumas & Nessimian (2014)* described *Oecetis iara*, the only known species so far in the Neotropics with the honeycomb texture on abdominal terga. However, in the Nearctic region, there are species that share this morphological characteristic such as *O. cinerascens* (Hagen) and *O. persimilis* (Banks).

Significant contributions on the taxonomy of *Oecetis* in the Neotropical region were made in the last 40 years by several authors (*Flint, 1974*; *Bueno-Soria, 1981*; *Rueda-Martín, Gibon & Molina, 2011*; *Blahnik & Holzenthal, 2014*; *Henriques-Oliveira, Dumas & Nessimian, 2014*; *Quinteiro & Calor, 2015*). The species proposed by those authors account for more than 50% of the currently known species in the Neotropical region, but it is known that there is much more to do in caddisfly taxonomy in the Neotropics (*Holzenthal & Calor, 2017*). Despite this increasing description of the Neotropical fauna, much of the region's biodiversity likely still remains unknown. There are species already deposited and labeled in museums waiting to be described. This study advances the knowledge of Neotropical caddisfly diversity by describing fourteen new species of *Oecetis* based on morphological characteristics of the adult male.

## MATERIALS AND METHODS

The specimens were primarily collected by use of ultraviolet fluorescent light bulbs placed in front of a white sheet, pan light traps (*Calor & Mariano, 2012*), and Malaise traps. Those specimens collected on the white sheets are preserved dried and pinned. The remaining specimens were preserved in 80% ethyl alcohol.

For a more accurate study of some genital characters, genitalia were removed and cleared in 85% lactic acid (*Blahnik, Holzenthal & Prather, 2007*) at 115 °C for approximately 1 h, washed with distilled water, and stored in 0.2 mL vials in approximately 50 μL of glycerin in the vial bottom.

The specimens were examined and illustrated with the aid of a stereomicroscope with drawing tube attached. Pencil sketches were scanned with a flat-bed scanner, and placed in Adobe Illustrator CS5, where they were digitally inked. Species descriptions were made using the DELTA editor (*Dallwitz, Paine & Zurcher, 1999*). Numbers in parentheses after "forewing length" in each description represent the number of specimens that integrate the type series and were used to calculate the average forewing length.

Terminology for wing venation and male morphology follows *Quinteiro & Calor (2015)*. Type specimens are deposited at the University of Minnesota Insect Collection, St. Paul, Minnesota, USA (UMSP), National Museum of Natural History, Smithsonian Institution, Washington, DC, USA (NMNH), Muzeu de Zoologia da Universidade de São Paulo, São Paulo, Brazil (MZSP), Museo de Historia Natural Noel Kempff Mercado, Santa Cruz de la Sierra, Bolivia (UASC), Colección Entomológica de la Universidad de Antioquia, Medellín, Colombia (CEUA), and Museu de Zoologia da Universidade Federal da Bahia, Salvador, Brazil (UFBA), as indicated in the material examined. Distribution for

each species is given by country and state, province, or department, summarized by the map presented at the end of the descriptions. The map was built using the website SimpleMappr (available at http://www.simplemappr.net). Species distributions are available as a Supplemental Data File (.kml) and can be opened in Google Earth. For those specimens with collection labels that did not include geographical coordinates, approximate coordinates were used to plot into the map based on the other label data.

The electronic version of this article in portable document format will represent a published work according to the International Commission on Zoological Nomenclature (ICZN), and hence the new names contained in the electronic version are effectively published under that Code from the electronic edition alone. This published work and the nomenclatural acts it contains have been registered in ZooBank, the online registration system for the ICZN. The ZooBank LSIDs (Life Science Identifiers) can be resolved and the associated information viewed through any standard web browser by appending the LSID to the prefix http://zoobank.org/. The LSID for this publication is: *urn:lsid: zoobank.org:pub:ED02CA58-B074-45A6-AAC7-48FB48B97BA8*. The online version of this work is archived and available from the following digital repositories: PeerJ, PubMed Central and CLOCKSS.

## Taxonomy

*Oecetis acuticlasper* Quinteiro & Holzenthal, n. sp. *urn:lsid:zoobank.org:act:046E520D-07ED-4892-BBDE-BE0654C5BE95*

**Diagnosis.** This species can be distinguished from all other *Oecetis* by the slender dorsal lobe of tergum X, which is almost as long as the preanal appendage and laterally divided at its apex; and by the shape of the inferior appendage, with its enlarged, somewhat triangular ventral lobe, discrete, also triangular dorsal lobe, and a distal lobe, which is distinctly constricted and narrowed in the apical third of the appendage.

This species is morphologically similar to *O. maspeluda Botosaneanu, 1977*. Both of them have a dorsoventrally divided tergum X, short phallic apparatus, and a somewhat triangular inferior appendage. However, the dorsal lobe of tergum X in this new species is much shorter than in *Oecetis maspeluda* and laterally divided at the apex. Also, the new species has the ventral margin of inferior appendage slightly concave in lateral view, giving an almost 90° angle between the ventral and distal lobes, while in *Oecetis maspeluda* this margin is smooth and almost straight. Additionally, the constriction present at the last third of the inferior appendage's length in the new species is unique in *Oecetis*. This species does not have features to place it in any known species group.

**Description. Male:** forewing length 6.5 mm ($n = 1$).

**Head.** Color yellowish brown (specimens in alcohol); maxillary palps pale yellow, 5-segmented, palpomeres subequal, densely covered with setae; labial palps yellow, 3-segmented.

**Thorax.** Pterothorax yellowish brown; forewing yellow; dark bands over cord absent, dark spots absent (Fig. 1A); fork I petiolate, fork V rooted; sectoral crossvein (*s*) not aligned

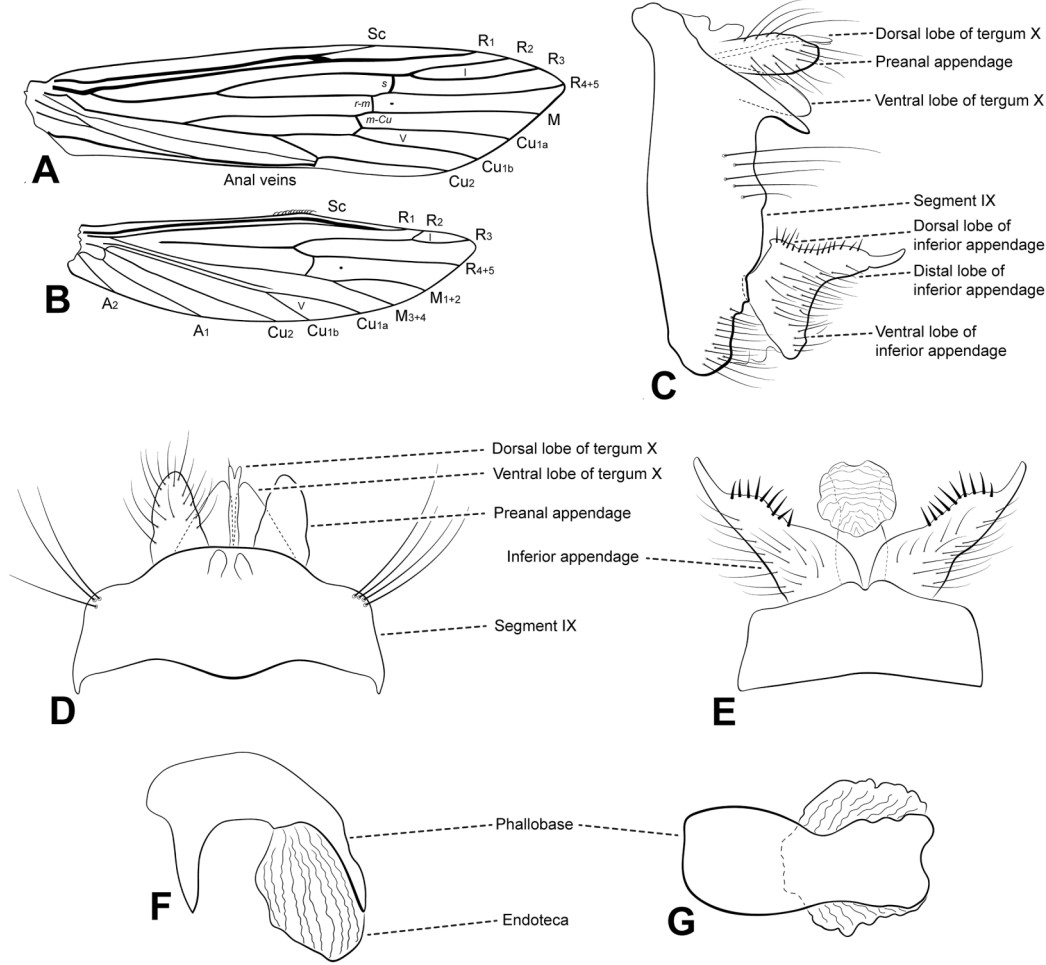

**Figure 1 Male genitalia of *Oecetis acuticlasper* n. sp.** *Oecetis acuticlasper* n. sp., Quinteiro & Holzenthal, male holotype. (A) forewing. (B) hindwing. (C) genitalia, lateral view. (D) genitalia, dorsal view. (E) genitalia, ventral view. (F) phallic apparatus, lateral view. (G) phallic apparatus, dorsal view.

with *r-m* (Fig. 1A). Hind wing with forks I and V present (Fig. 1B). Legs yellowish brown, mid leg with longitudinal row of spines on tibia and tarsal segments. Tibial spur formula 0,2,2.

**Abdomen.** Segment IX annular, short, bearing pair of lateral processes, slender, straight, tapering posteriorly, shorter than phallic apparatus; acrotergite absent (Figs. 1C and 1D). Preanal appendage short, digitate, bearing apical setae (Figs. 1C and 1D). Tergum X, in lateral view, divided into dorsal and ventral lobes (Fig. 1C); dorsal lobe modified into single cylindrical structure, with apex slightly divided, longer than ventral lobe, and with short apical setae (Figs. 1C and 1D); ventral lobe divided medially, with V-shaped incision, broad basally, digitate apically (Figs. 1C and 1D). Inferior appendage 1-segmented, setose; dorsal lobe acuminate, discrete (Figs. 1C and 1E); ventral lobe protruding, triangular, apex acute; distal lobe narrow, tapering posteriorly, apex acute, with distinct constriction at distal third (Figs. 1C and 1E); short, stout spine-like setae present on inner surface (Fig. 1E). Phallic apparatus bilaterally symmetrical, cylindrical,

short, strongly curved ventrally, with globular endotheca (Figs. 1F and 1G). Phallic spine and phallotremal sclerite absent (Figs. 1F and 1G).

**Distribution.** Brazil (Minas Gerais, Rio de Janeiro).

**Material examined. Holotype: (male):** BRAZIL, **Minas Gerais**, Presidente Olegário, Faz. Gigante, Armadilha Luminosa, 18°31′S, 46°18′W, 1,000 m, 02–05.iv.2007, Amorim, Ribeiro, Capellari, Borkent (MZSP). **Paratype: BRAZIL, Rio de Janeiro**, Nova Friburgo, mun. water supply, 950 m, 24 April 1977, C.M. & O.S. Flint Jr.—one male (NMNH; identified as *Oecetis* n. sp. E in loan to UMSP).

**Etymology.** From Latin *acutus* = pointed, in reference to the pointed tip of the inferior appendage, or clasper.

*Oecetis flinti* Quinteiro & Holzenthal, n. sp. *urn:lsid:zoobank.org:act:E760A8EB-7908-427C-AC19-D11291E15FE8*

**Diagnosis.** This species can be diagnosed from other *Oecetis* by its enlarged tergum IX, ovoid preanal appendage, and by the inferior appendage, which has a basal constriction and lacks dorsal and ventral lobes.

Important adjunct characters, such as the presence of dark spots on the forewing and the position of the main forks, are easily observable. This new species is similar to *O. pratti Denning, 1947* since both have tergum X divided dorsoventrally, with a deflexed dorsal lobe, and an inferior appendage without dorsal and ventral lobes. However, the inferior appendage of this new species has a distinct constriction at its base, while *Oecetis pratti* has the inferior appendage uniformly wide along its entire length. *Oecetis flinti* n. sp. has its preanal appendage short and ovoid, somewhat lobulate in lateral view, while in *Oecetis pratti* they are long and digitate. The most evident characteristics of this new species that differ from *Oecetis pratti* are the presence of a phallic spine in the phallic apparatus and the elongate tergum IX, which is much longer than sternum IX. This species does not fit within one of the recognized species groups.

**Description. Male:** forewing length 4.5 mm ($n = 1$).

**Head.** Color yellowish brown (specimens in alcohol); maxillary palps pale yellow; labial palps pale yellow, 3-segmented.

**Thorax.** Pterothorax yellowish brown; forewing yellow; dark bands over cord present (Fig. 2A); dark spots on M-Cu fork, on basis of Rs, on basis of $Cu_1$ and $Cu_2$, on junction of anal veins (Fig. 2A); forks I and V rooted; sectoral crossvein (*s*) not aligned with *r-m* (Fig. 2A). Hind wing with forks I and V present (Fig. 2B). Legs pale yellow, mid leg with longitudinal row of spines on tibia and tarsal segments. Tibial spur formula 1,2,2; apical spur of fore tibia very small.

**Abdomen.** Segment IX uneven, tergum IX longer than sternum IX; acrotergite absent (Fig. 2C). Preanal appendage slightly wider than long (ovoid), bearing apical setae (Figs. 2C and 2D). Tergum X, in lateral view, not divided into dorsal and ventral lobes, undivided medially, composed of single elongated lobe, broad basally, tapering apically, apex acuminate (Figs. 2C and 2D). Inferior appendage 1-segmented, setose, broad basally;
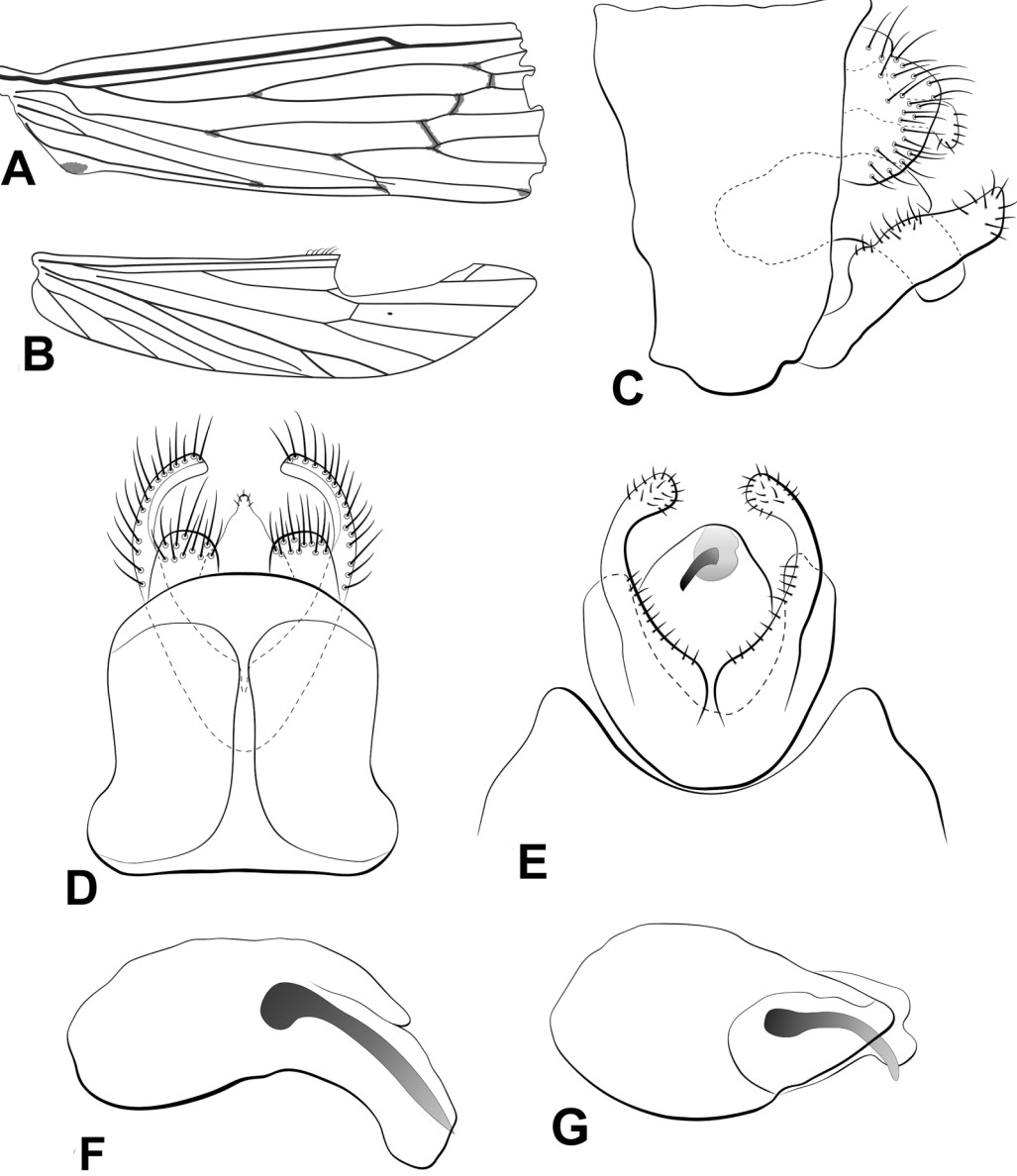

**Figure 2 Male genitalia of *Oecetis flinti* n. sp.** *Oecetis flinti* n. sp., Quinteiro & Holzenthal, male holotype. (A) forewing. (B) hindwing. (C) genitalia, lateral view. (D) genitalia, dorsal view. (E) genitalia, ventral view. (F) phallic apparatus, lateral view. (G) phallic apparatus, dorsal view.

ventral and dorsal lobes absent (Figs. 2C–2E); distal lobe narrow, tapering posteriorly, apex acute, discrete constriction close to base, slightly bent inward (Figs. 2C–2E); short stout spine-like setae absent (Figs. 2C–2E). Phallic apparatus slightly bilaterally asymmetrical, curved downward, cylindrical, elongate, rounded on base, tapering apically, apex acute (Figs. 2F and 2G). Phallotremal sclerite absent (Figs. 2F and 2G).

**Distribution.** Brazil (Tocantins).

**Material examined. Holotype (male): BRAZIL, Tocantins,** Mateiros, P.E. Jalapão, Cachoeira da Formiga, 10°20′58″S, 46°28′23.1″W, 461 m, 17.x.2008, luz, Calor, Mariano, Mateus (MZSP). **Paratypes:** same data as holotype—two males (UFBA).

**Taxonomic comment:** The available specimens had damaged wings so it was not possible to illustrate an entire wing.

**Etymology.** The specific epithet honors our colleague Dr. Oliver S. Flint for his contributions to caddisfly taxonomy and his extensive studies in the Neotropical region.

*Oecetis carinata* Quinteiro & Holzenthal, n. sp. *urn:lsid:zoobank.org:act:404BC99D-A18C-4322-892F-E824DA3B66CF*

**Diagnosis.** This species can be differentiated from other *Oecetis* by its laterally deeply divided tergum X, forming two slender terete lobes, and its inferior appendage with the ventral lobe quadrate and greatly enlarged, and its distal lobe rounded and very short.

This new species is very similar to *O. inflata* Flint, 1974 since they both share a deeply divided tergum X, forming two slender and terete processes. Additionally, they resemble each other in that the preanal appendages are short and digitate, the phallic apparatus is short and strongly curved ventrally, and the forewing venation is almost identical. However, the new species has the ventral lobe of the inferior appendage quadrate and greatly enlarged in relation to the distal lobe. *Oecetis inflata* has the ventral lobe of the inferior appendage somewhat quadrate, but the ventral margin is very smooth and not as angled as in the new species. Additionally, the distal lobe of the inferior appendage in *Oecetis carinata* is reduced compared to *Oecetis inflata*. This new species does not have characteristics that allow placement in a known species group.

**Description. Male:** forewing length 6.5 mm ($n = 1$).

**Head.** Color yellowish brown (specimens in alcohol); maxillary palps yellowish brown, 5-segmented, palpomeres subequal; labial palps pale yellow, 3-segmented.

**Thorax.** Pterothorax yellow; forewing yellowish brown; dark bands over cord absent (Fig. 3A); dark spots absent; forks I and V rooted (Fig. 3A); sectoral crossvein (*s*) aligned with *r-m* (Fig. 3A). Hind wing with forks I and V present (Fig. 3B). Legs yellowish brown, mid leg with longitudinal row of spines over distal half of femur, all along tibia and first tarsal segment. Tibial spur formula 0,2,2.

**Abdomen.** Segment IX annular, short; acrotergite absent (Fig. 3C). Preanal appendage short, digitate, bearing apical setae (Figs. 3C and 3D). Tergum X, in lateral view, divided in dorsal and ventral lobes; dorsal lobe divided medially, forming two slender, terete lobes, apices acuminate; ventral lobe undivided, short, apex rounded in dorsal view and acute in lateral view (Figs. 3C and 3D). Inferior appendage 1-segmented, setose; dorsal lobe absent (Figs. 3C and 3E); ventral lobe quadrate, keeled, enlarged, margin angular (Figs. 3C and 3E); distal lobe a broad, smoothly rounded projection; short and stout spine-like setae absent (Figs. 3C and 3E). Phallic apparatus bilaterally symmetrical, cylindrical, short, strongly curved ventrally (Figs. 3F and 3G). Phallic spine and phallotremal sclerite absent (Figs. 3F and 3G).

**Distribution.** Brazil (Bahia, São Paulo).

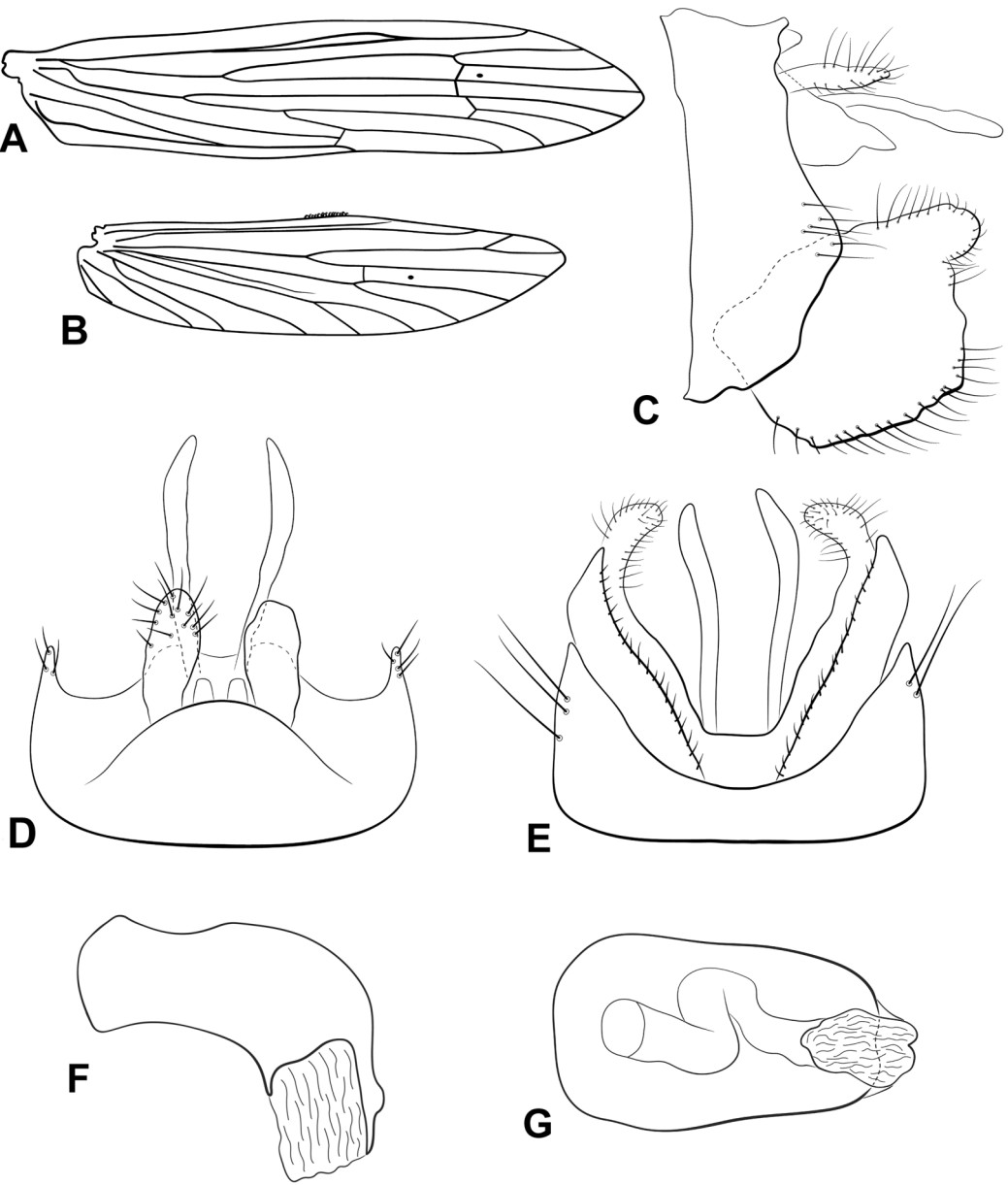

**Figure 3 Male genitalia of *Oecetis carinata* n. sp.** *Oecetis carinata* n. sp., Quinteiro & Holzenthal, male holotype. (A) forewing. (B) hindwing. (C) genitalia, lateral view. (D) genitalia, dorsal view. (E) genitalia, ventral view. (F) phallic apparatus, lateral view. (G) phallic apparatus, dorsal view.

**Material examined. Holotype (male):** BRAZIL, **São Paulo**, São Carlos, Córrego Fazzari, 20.xii.2007, malaise trap (MZSP). **Paratypes: BRAZIL, Bahia**, Mucugê, Sempre Viva, Córrego do Boiadeiro, 10.v.2015, malaise, Dias & Campos—two males, two females (UFBA).

**Etymology.** From Latin *carina* = keel, referring to the keel-like ventral lobe of the inferior appendage in ventral view.

*Oecetis cassicoleata* Quinteiro & Holzenthal, n. sp. *urn:lsid:zoobank.org:act:140E12D6-B022-4128-9AB9-BF148264378F*

**Diagnosis.** This species can be distinguished from the other species in *Oecetis* by its inferior appendage, which has the distal lobe curved, forming an obtuse angle along its length, and by the enlarged phallobase, with a lamellate process distally over the endotheca.

This new species has no similar species in the Neotropical region. Some species, such as *O. testacea* (*Curtis, 1834*) or *O. cinerascens* (*Hagen, 1861*), share an enlarged phallic apparatus, but both of them have phallic spines and a modified, reticulate tergum VIII, both characteristics not present in this species. Also, the inferior appendage of the new species is very distinct, having a rounded, slightly projected dorsal lobe, which becomes forward-pointing due to the concavity of the distal lobe, which forms an obtuse angle. Additionally, the presence of a flap-like projection on the dorsodistal margin of the phallobase is very conspicuous. This is another new species that cannot be placed in a named species group.

**Description. Male:** forewing length 5.5 mm ($n = 1$).

**Head.** Color yellowish brown (specimens in alcohol). Scape stout; pedicel short; maxillary palps pale yellow, 5-segmented; labial palps pale yellow, 3-segmented.

**Thorax.** Pterothorax yellowish brown; forewing yellow; dark bands over cord absent (Fig. 4A); dark spots on forks, junctions and end of veins; fork I rooted, fork V sessile; sectoral crossvein (*s*) not aligned with *r-m* (Fig. 4A). Hind wing with forks I and V present (Fig. 4B). Legs yellowish brown. Tibial spur formula 0,2,2.

**Abdomen.** Segment IX annular, short; acrotergite absent (Figs. 4C and 4D). Preanal appendage short, rounded, bearing apical setae (Figs. 4C and 4D). Tergum X, in lateral view not divided into dorsal and ventral lobes, composed of single elongate lobe, undivided medially, broad basally, tapering apically, apex acute (Figs. 4C and 4D). Inferior appendage 1-segmented, setose; in lateral view, dorsal lobe broad, rounded, projecting forward (Fig. 4C); ventral lobe absent (Figs. 4C and 4D); in lateral view, distal lobe narrow, cylindrical, bent posteriorly, forming a concavity ventrally, apex rounded (Fig. 4C); short, stout spine-like setae absent (Figs. 4C and 4D). Phallic apparatus slightly bilaterally asymmetrical, curved downward, cylindrical, elongate, broad basally, tapering distally, apex truncate, bearing flap-like projection distally on dorsal surface (Figs. 4E and 4F). Phallic spine and phallotremal sclerite absent (Figs. 4E and 4F).

**Distribution.** Brazil (Bahia).

**Material examined. Holotype (male): BRAZIL, Bahia,** Barreiras, Rio de Janeiro, cach. Acaba Vidas, 11°53′37″S, 45°36′09″W, alt. 722 m, 14.x.2008, light trap, Calor, Mariano, Mateus (MZSP).

**Etymology.** From Latin *cassis* = cap and *coleatus* = pertaining to the penis, referring to the hood-like projection on the dorsal surface of phallic apparatus covering the endotheca.

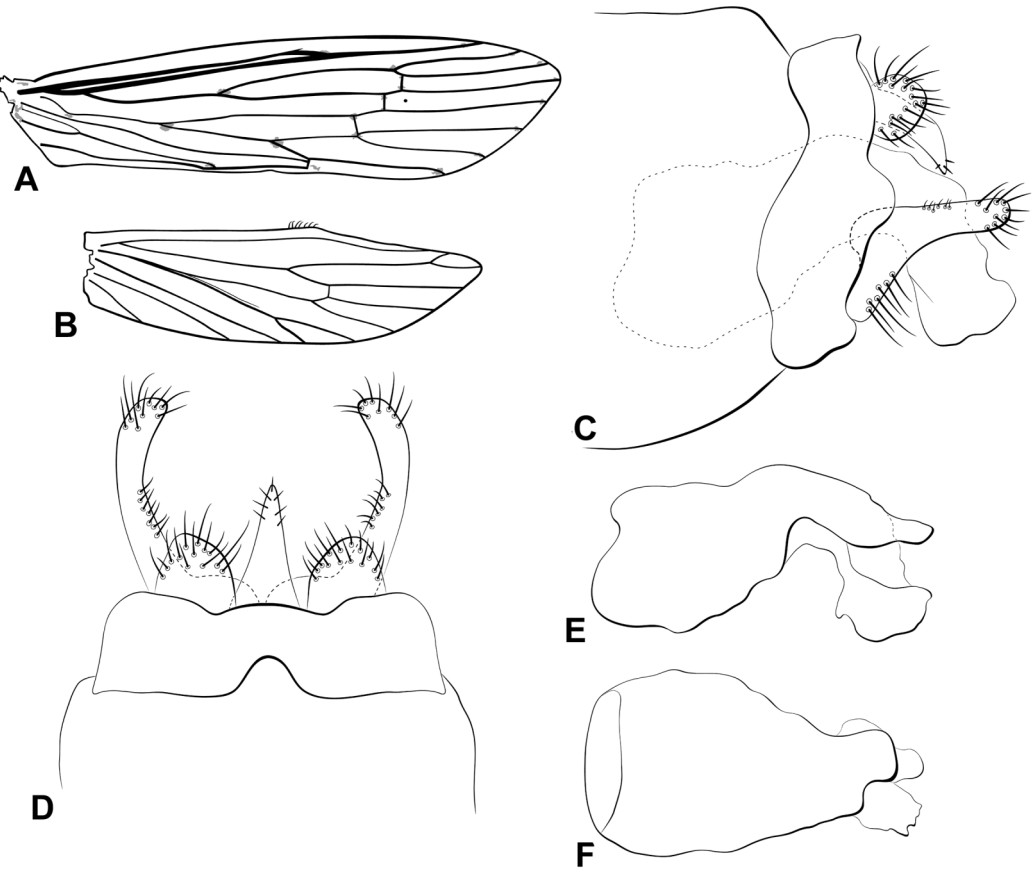

**Figure 4 Male genitalia of *Oecetis cassicoleata* n. sp.** *Oecetis cassicoleata* n. sp., Quinteiro & Holzenthal, male holotype. (A) forewing. (B) hindwing. (C) genitalia, lateral view. (D) genitalia, dorsal view. (E) phallic apparatus, lateral view. (F) phallic apparatus, dorsal view.

*Oecetis bidigitata* Quinteiro & Holzenthal, n. sp. *urn:lsid:zoobank.org:act:7A089FD8-F3F1-4339-898F-98DE402E3C81*

**Diagnosis.** This new species can be distinguished from the others in the *punctata* group by its stirrup-like phallotremal sclerite and by its inferior appendage, with a very short ventral lobe and two digitate projections bearing strong and thick setae on the distal lobe.

This species is very similar to *O. knutsoni* Flint, 1981 and *O. quasipunctata* n. sp. However, this species, differs from *Oecetis knutsoni* and *Oecetis quasipunctata* n. sp., by having a distinct stirrup-like phallotremal sclerite, not present in the other two species. Additionally, *Oecetis bidigitata* n. sp. has only two digitate projections bearing strong and thick setae on the apex of the distal lobe of inferior appendage, while *Oecetis knutsoni* and *Oecetis quasipunctata* n. sp. have four projections.

**Description. Male:** forewing length 7.5–8.5 mm ($n = 7$).

**Head.** Color pale yellow (pinned specimens). Antennae three times length of forewing; scape stout, elongate; pedicel enlarged in width, subequal to other flagellomeres in length, shorter than scape. Maxillary palps yellowish brown, 5-segmented, palpomeres subequal in length and width, setose. Labial palps pale yellow, 3-segmented.

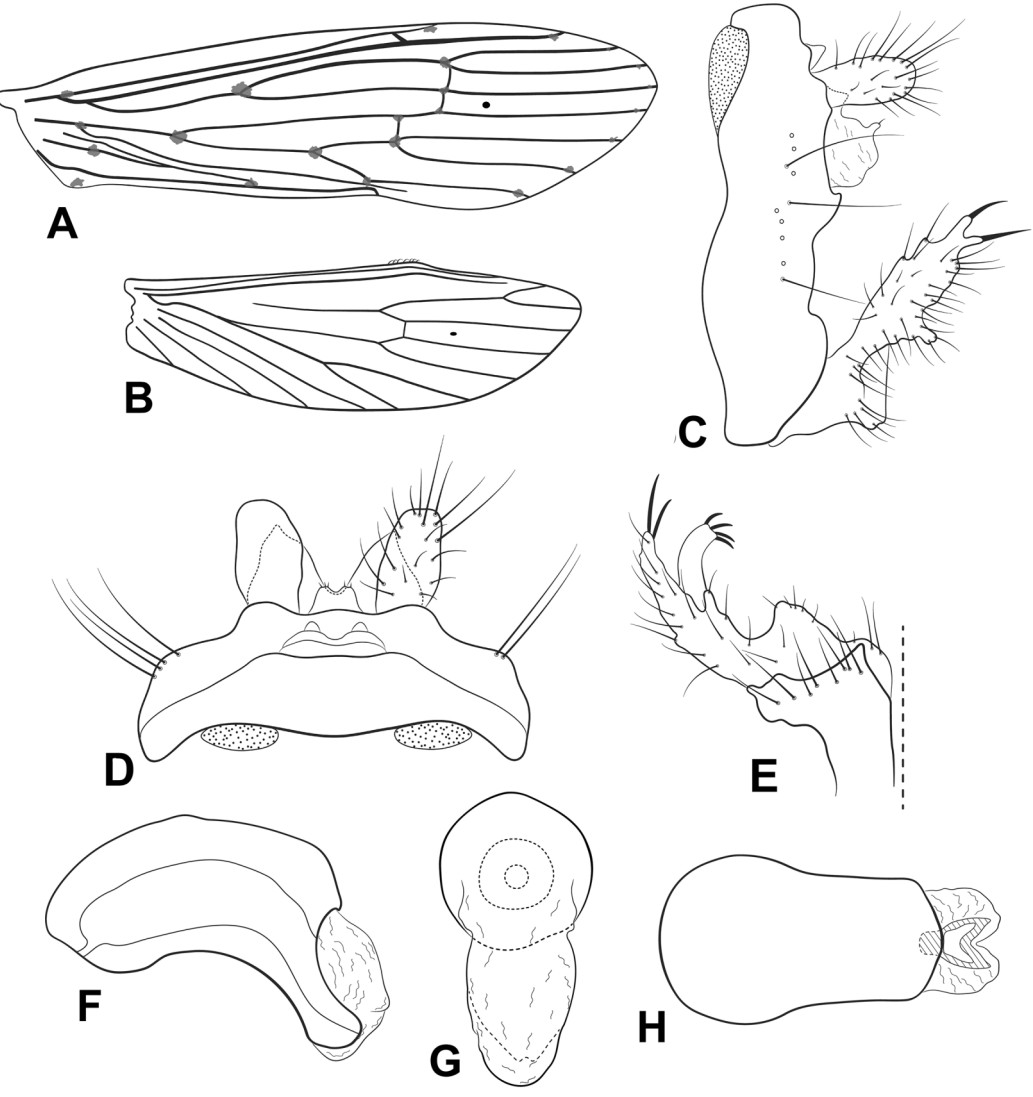

**Figure 5 Male genitalia of *Oecetis bidigitata* n. sp.** *Oecetis bidigitata* n. sp., Quinteiro & Holzenthal, male holotype. (A) forewing. (B) hindwing. (C) genitalia, lateral view. (D) genitalia, dorsal view. (E) inferior appendage, ventral view. (F) phallic apparatus, lateral view. (G) phallic apparatus, caudal view. (H) phallic apparatus, dorsal view.

**Thorax.** Pterothorax yellowish brown; forewing yellow; dark bands over cord absent (Fig. 5A); dark spots on forks, junctions and end of veins; fork I rooted, fork V sessile (Fig. 5A); sectoral crossvein (*s*) aligned with *r-m* (Fig. 5A). Hind wing with forks I and V present (Fig. 5B). Legs yellow, mid leg with longitudinal row of spines on tibia and tarsal segments. Tibial spur formula 1,2,2.

**Abdomen.** Segment IX annular, short; two acrotergites present dorsolaterally (Figs. 5C and 5D). Preanal appendage very short, digitate, bearing apical setae (Figs. 5C and 5D). Tergum X, in lateral view not divided in dorsal and ventral lobes, divided medially by V-shape incision, in dorsal view broad basally, acute apically (Figs. 5C and 5D). Inferior appendage 1-segmented, broad basally, setose; dorsal lobe absent (Figs. 5C and 5E);

ventral lobe quadrate, short (Figs. 5C and 5E); distal lobe narrow, short, two digitate projections on top, inner process curved inward (Figs. 5C and 5E); two short, stout spine-like apical setae present (Figs. 5C and 5E). Phallic apparatus bilaterally symmetrical, cylindrical, elongate, membranous apically, curved ventrally (Figs. 5F–5H); in caudal view, phallobase narrowing distally, somewhat U-shaped (Fig. 5G). Phallic spine absent (Figs. 5F–5H). Phallotremal sclerite U-shapped, with V-shaped incision distally (Fig. 5H).

**Distribution.** Bolivia (La Paz).

**Material examined. Holotype (male):** **BOLÍVIA, Dept. La Paz,** San Buenaventura-Ixiamas rd., Arroyo Maije at Puente Maije, 14°20.908′S, 67°40.530′W, 278 m, 14.vii.2003, Robertson and Blahnik (UASC, on loan to UMSP). **Paratypes:** same data as holotype—one female (UMSP); **BOLÍVIA, Dept. La Paz,** ANMI Madidi, Chalalan Ecolodge, Rio Tuichi at entrance to lodge & trib., 14°25.017′S, 67°54.378′W, 300 m, 27.vii.2003, Robertson and Blahnik—three males (UMSP); same data except Raya Mayo river at Anta trail, 14°28.134′S, 67°55.761′W, 264 m, 26.vii.2003, Robertson and Blahnik—three males (NMNH).

**Etymology.** From Latin prefix *bi* = two and *digitus* = finger, referring to the two digitate projections on apex of the inferior appendages.

*Oecetis quasipunctata* Quinteiro & Holzenthal, n. sp. *urn:lsid:zoobank.org:act:B7E84B92-234B-46F1-BDF0-9D74D1CA9AB9*

**Diagnosis.** This new species can be distinguished from the others in the *punctata* group by its inferior appendage, with the ventral lobe very reduced and forming an acute angle with the distal lobe, and also by the nearly straight distal lobe.

This new species is very similar to *Oecetis knutsoni, Oecetis bidigitata* n. sp., and *Oecetis punctata* (*Navás, 1924*) (Fig. 7). *Oecetis bidigitata* has only two digitate projections apically on the distal lobe of the inferior appendage, while *Oecetis knutsoni, Oecetis punctata*, and *Oecetis quasipunctata* n. sp. have four projections. *Oecetis punctata* has a concave inner surface of the inferior appendage (Fig. 7E, best seen in ventral view), while *Oecetis knutsoni, Oecetis bidigitata* n. sp., and *Oecetis quasipunctata* n. sp. have the inner surface straight (Fig. 6E). The main differences between *Oecetis knutsoni* and this new species are in the inferior appendage. *Oecetis knutsoni* has an inferior appendage with a long ventral lobe forming a straight angle with the distal lobe, while this new species has a very reduced ventral lobe forming an acute angle with the distal lobe. Additionally, *Oecetis knutsoni* has the apex of the distal lobe slightly bent posteriorly, while the new species has it nearly straight.

**Description. Male:** forewing length 9.1–10.3 mm (*n* = 21).

**Head.** Color yellowish brown (pinned specimens). Antennae three times length of forewing; scape stout, elongate; pedicel enlarged in width, subequal to other flagellomeres in length, shorter than scape. Maxillary palps yellow, 5-segmented, palpomeres subequal in length and width. Labial palps yellow, 3-segmented.

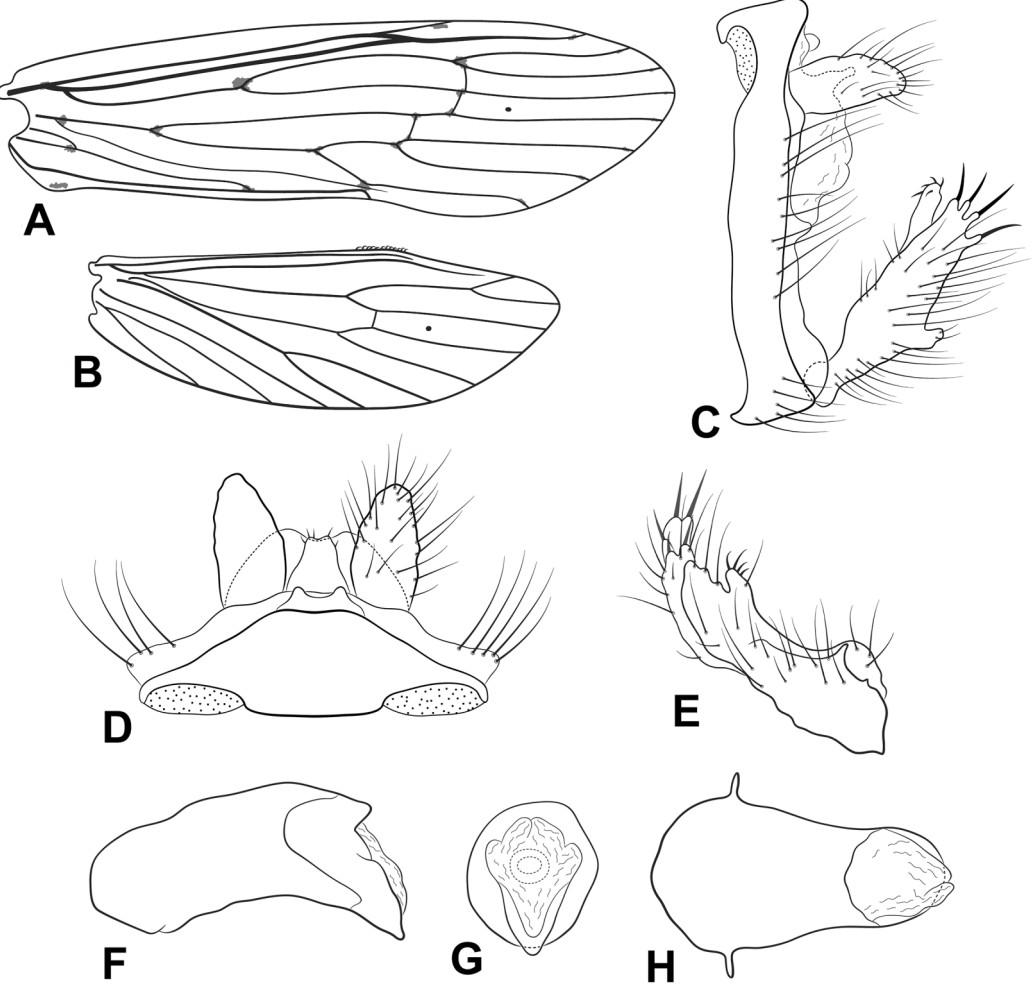

**Figure 6 Male genitalia of *Oecetis quasipunctata* n. sp.** *Oecetis quasipunctata* n. sp., Quinteiro & Holzenthal, male holotype. (A) forewing. (B) hindwing. (C) genitalia, lateral view. (D) genitalia, dorsal view. (E) inferior appendage, ventral view. (F) phallic apparatus, lateral view. (G) phallic apparatus, caudal view. (H) phallic apparatus, dorsal view.

**Thorax.** Pterothorax yellow; forewing yellow; dark bands over cord absent (Fig. 6A); dark spots on forks, junctions and end of veins; forks I and V rooted (Fig. 6A); sectoral crossvein (*s*) aligned with *r-m* (Fig. 6A). Hind wing with forks I and V present (Fig. 6B). Legs pale yellow, mid leg with longitudinal row of spines on tibia and tarsal segments. Tibial spur formula 1,2,2.

**Abdomen.** Segment IX annular, very short; two acrotergites present dorsolaterally (Fig. 6C). Preanal appendage short, digitate, bearing apical setae (Figs. 6C and 6D). Tergum X, in lateral view, not divided into dorsal and ventral lobes, composed of single elongate lobe, broad basally, divided mesally by shallow depression (Figs. 6C and 6D). Inferior appendage 1-segmented, broad basally, setose (Figs. 6C and 6E); dorsal lobe absent; ventral lobe quadrate, very short (Figs. 6C and 6E); distal lobe narrow, straight, short, with four digitate projections on top, inner process curved inward; short and stout spine-like apical setae present (Figs. 6C and 6E). Phallic apparatus bilaterally symmetrical,

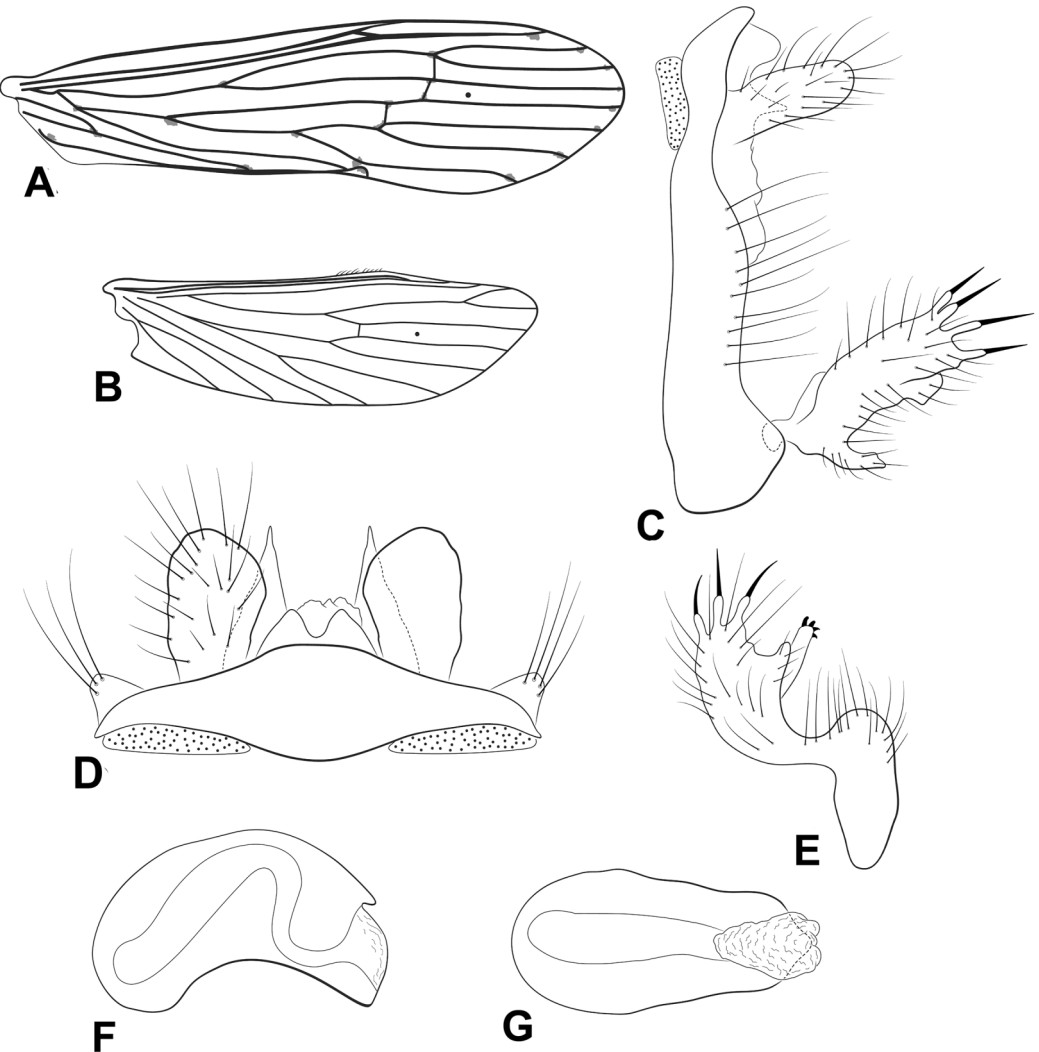

**Figure 7 Male genitalia of *Oecetis punctata* (*Navás, 1924*).** *Oecetis punctata* (*Navás, 1924*); specimen label: Colombia, Magdalena, Municipio de Santa Maria, Rio Minca en Minca, 11°08′35″N, 74°06′58″W, 570 m, 09.xii.1997, Muñoz-Quesada et al. (A) forewing. (B) hindwing. (C) genitalia, lateral view. (D) genitalia, dorsal view. (E) inferior appendage, ventral view. (F) phallic apparatus, lateral view. (G) phallic apparatus, dorsal view.

curved downward, cylindrical, short (Figs. 6F–6H); in caudal view, apex short, cylindrical, phallobase tapering distally, with slightly "pinched" tip (Fig. 6G). Phallic spine and phallotremal sclerite absent (Figs. 6F–6H).

**Distribution.** Colombia (Cauca, Valle, Quindió).

**Material examined. Holotype (male): COLOMBIA, Cauca**, Municipio de Inzá, Quebrada San Andrés, 1 km S del centro de San Andrés de Pisimbalá, 02°34′36″N, 76°02′11″W, 1,730 m, 20.xii.1997, Fdo. Muñoz-Q. et al. (CEUA, on loan to UMSP); **Paratypes:** same data as holotype—14 males (UMSP); **COLOMBIA, Cauca**, Municipio de Inzá, 500 m W Restaurante "La Portada", San Andrés de Pisimbalá, 02°34′56″N, 76°02′36″W, 1,750 m, 21.xii.1997, Fdo. Muñoz-Q. et al.—16 males (UMSP);

Municipio de Belalcazar, Quebrada Tálaga, ~14 km N Páez (Belalcazar), 02°42′24″N, 76°01′05″W, 1,680 m, 22.xii.1997, Fdo. Muñoz-Q. et al.—12 males (UMSP); Río Cabuyal Pescador, ~20 km N Piendamó (Carretera Panamericana), 02°48′N, 76°32′W, 1,400 m, 28.xii.1997, Fdo. Muñoz-Q. et al.—four males (CEUA, on loan to UMSP); **Valle**, Municipio El Cerrito, Río Cerrito, 7.1 km E Hacienda "El Paraíso", 03°38′59″N, 76°09′10″W, 1,950 m, 03.xii.1997, Fdo. Muñoz-Q. et al.—four males, two females (UFBA); Municipio de Cali, Río Pichindé, Peñas Blancas, ~24 km SW Cali, 03°25′06″N, 76°39′04″W, 2,000 m, 18.xii.1997, Fdo. Muñoz-Q. et al.—12 males, two females (UMSP); **Quindió**, Río Quindió, Retén "La Playa", ~2 km NE Salento, 04°38′25″N, 75°33′24″W, 1,890 m, 02.i.1998, Fdo. Muñoz-Q. et al.—two males, two females (NMNH).

**Etymology.** From Latin *quasi* = appearing as if, similar. This is a reference to the resemblance of this new species with *Oecetis punctata* (*Navás, 1924*), illustrated in Fig. 7.

*Oecetis calori* Quinteiro & Holzenthal, n. sp. *urn:lsid:zoobank.org:act:58B08D3F-32E2-4408-9D62-24A46FAB2B5F*

**Diagnosis.** This new species can be diagnosed from the others in the *falicia* group by its three phallic spines and by the inferior appendage with a small, discrete, quadrate basal projection with a truncate apex, and a triangular, protruding, and very angular ventral lobe.

This new species is similar to *O. fibra* Chen & Morse *in Quinteiro & Calor, 2012* and to *O. acarati Angrisano & Sganga, 2009*; all of them share a segment IX with slender, ventrally directed dorsolateral processes with no ramifications, and a terete, apically rounded distal lobe on the inferior appendage. However, none of the described species possess three spines in the phallic apparatus as in the new species. Additionally, the inferior appendage of *Oecetis fibra* has a quadrate ventral lobe with smooth margins, while in *Oecetis acarati* and the new species this lobe is triangular. *Oecetis acarati* has the ventral lobe of inferior appendage with a smooth margin and no dorsal lobe, while *Oecetis calori*, n. sp. has a discrete quadrate dorsal lobe and a triangular flattened ventral lobe.

**Description. Male:** forewing length 7.3–7.6 mm ($n = 3$).

**Head.** Color yellowish brown (pinned specimens). Antennae three times length of forewing; scape stout, elongate; pedicel enlarged in width, subequal to other flagellomeres in length, shorter than scape. Maxillary palps yellow, 5-segmented. Labial palps yellow, 3-segmented.

**Thorax.** Pterothorax yellow; forewing yellowish brown; dark bands over cord absent (Fig. 8A); dark spots absent; forks I and V rooted (Fig. 8A); sectoral crossvein (*s*) not aligned with *r-m* (Fig. 8A). Hind wing with forks I and V present (Fig. 8B). Legs pale yellow, mid leg with longitudinal row of spines on tibia and tarsal segments. Tibial spur formula 1,2,2.

**Abdomen.** Segment IX annular, short, bearing pair of dorsolateral processes; processes slender, bent ventrad, cylindrical, tapering posteriorly, same length as phallic apparatus; acrotergite absent (Figs. 8C and 8D). Preanal appendage long, digitate, bearing apical

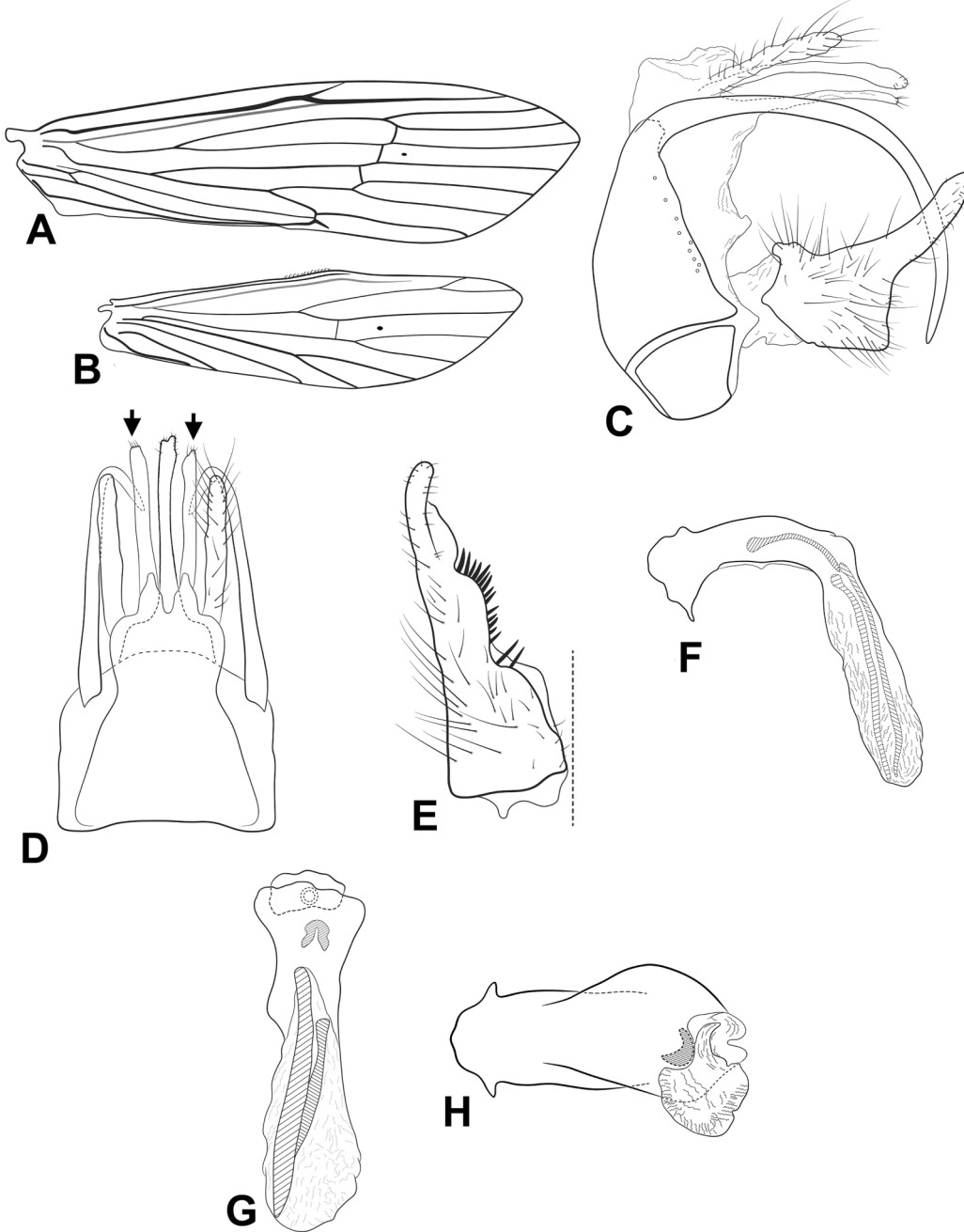

**Figure 8 Male genitalia of *Oecetis calori* n. sp.** *Oecetis calori* n. sp., Quinteiro & Holzenthal, male holotype. (A) forewing. (B) hindwing. (C) genitalia, lateral view. (D) genitalia, dorsal view. (E) inferior appendage, ventral view. (F) phallic apparatus, lateral view. (G) phallic apparatus, caudal view. (H) phallic apparatus, dorsal view. Black arrowheads indicate tergum X lobules.

setae (Figs. 8C and 8D). Tergum X, in lateral view, divided into dorsal and ventral lobes; dorsal lobe modified into single cylindrical structure, apex digitate, same length as ventral lobe, with short apical setae (Figs. 8C and 8D); ventral lobe deeply divided laterally, forming two cylindrical lobules, apices rounded (Figs. 8C and 8D, black arrowheads). Inferior appendage 1-segmented, broad at base, setose; dorsal lobe in lateral view

quadrate, discrete (Figs. 8C and 8E); ventral lobe, in lateral view, quadrate (Figs. 8C and 8E); distal lobe narrow, tapering posteriorly, apex rounded (Figs. 8C and 8E); short, stout spine-like setae present on inner surface (Fig. 8E). Phallic apparatus bilaterally symmetrical, curved downward, cylindrical, elongate, membranous apically (Figs. 8F–8H); apex elongate, in caudal view (Fig. 8G). Three phallic spines present, curved downward (Fig. 8F and 8G). One phallotremal sclerite present, horseshoe-shaped, with discrete concavities on sides (Figs. 8G).

**Distribution.** Brazil (Minas Gerais).

**Material examined. Holotype (male): BRAZIL, Minas Gerais,** Aldeia da Cachoeira das Pedras, 20°06.824′S, 44°01.412′W, 925 m, 28–29.ix.2000, Paprocki and Braga (MZSP). **Paratypes:** same data as holotype—one female (UMSP); **Brazil, Minas Gerais,** Estação Ecológica de Peti, Córrego Brucutu, 19°52.995′S, 43°22.452′W, 29.ix.1998, Paprocki—two males, one female (UMSP).

**Etymology.** The specific epithet honors our colleague Adolfo R. Calor for his contributions to caddisfly taxonomy and systematics, especially of the Brazilian fauna.

*Oecetis hastapulla* Quinteiro & Holzenthal, n. sp. *urn:lsid:zoobank.org:act:B43E4DCE-3579-4BDD-B803-B32151AB0327*

**Diagnosis.** This new species can be discriminated from the others in the *falicia* group by the very elongate dorsal lobe of tergum X, with its slightly clavate apex, the divided apex of the inferior appendage, and by the elongate, asymmetrical, dorsolateral processes on segment IX.

This new species is very similar to *O. prolongata* Flint, 1981, since both of them have an elongate and terete dorsal lobe of tergum X, an inferior appendage with the apex divided, and the dorsolateral processes of segment IX asymmetrical. However, this new species has the apex of dorsal lobe of tergum X clavate, while *Oecetis prolongata* has it uniform in width along its entire length. Also, *Oecetis hastapulla* n. sp. has the dorsolateral processes of segment IX longer than those of *Oecetis prolongata* and the process on the right side of the body is bent posteriorly, while those of *Oecetis prolongata* are bent ventrally.

**Description. Male:** forewing length 6.5 mm ($n = 2$).

**Head.** Color yellowish brown (pinned specimens). Antennae 2.5 times length of forewing; scape stout, elongate; pedicel enlarged in width, subequal to other flagellomeres in length, shorter than scape. Maxillary palps yellow, 5-segmented, setose. Labial palps pale yellow, 3-segmented.

**Thorax.** Pterothorax yellowish brown; forewing yellow; dark bands over cord absent (Fig. 9A); dark spots absent (Fig. 9A); forks I and V rooted; sectoral crossvein (*s*) not aligned with *r-m* (Fig. 9A). Hind wing with forks I and V present (Fig. 9B). Legs yellow, mid leg with longitudinal row of spines on tibia and tarsal segments. Tibial spur formula 0,2,2.

**Abdomen.** Segment IX annular, short, bearing pair of dorsolateral processes, slender, bent ventrad, cylindrical, tapering posteriorly, not bilaterally symmetrical, with dark tips,

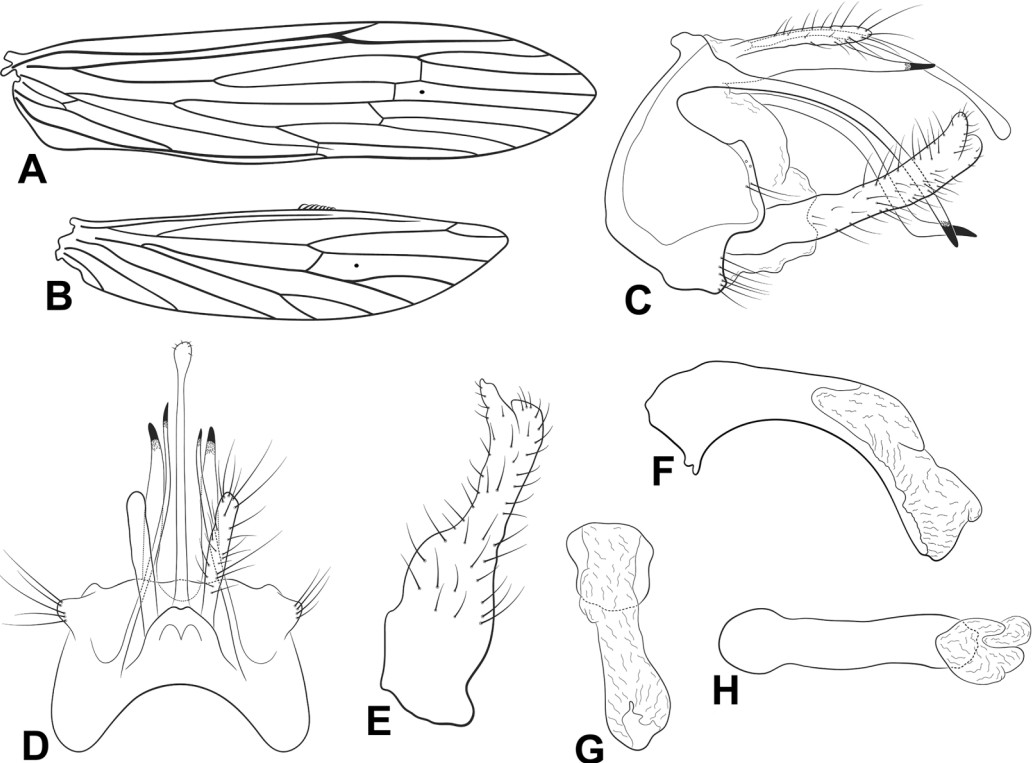

**Figure 9** **Male genitalia of *Oecetis hastapulla* n. sp.** *Oecetis hastapulla* n. sp., Quinteiro & Holzenthal, male holotype. (A) forewing. (B) hindwing. (C) genitalia, lateral view. (D) genitalia, dorsal view. (E) inferior appendage, ventral view. (F) phallic apparatus, lateral view. (G) phallic apparatus, caudal view. H phallic apparatus, dorsal view.

tip of right process bent 90 degrees distally, same length as phallic apparatus; acrotergite absent (Figs. 9C and 9D). Preanal appendage long, digitate, bearing apical setae (Figs. 9C and 9D). Tergum X, in lateral view, divided into dorsal and ventral lobes; dorsal lobe modified into single cylindrical structure, apex clavate, longer than ventral lobe, with short apical setae (Figs. 9C and 9D); ventral lobe deeply divided laterally, forming two cylindrical lobules, apex acuminate, tips dark (Figs. 9C and 9D). Inferior appendage 1-segmented, broad basally, setose (Figs. 9C and 9E); dorsal and ventral lobes absent (Fig. 9C); distal lobe narrow, tapering posteriorly, with V-shaped incision on apex (Figs. 9C and 9E); short, stout spine-like setae absent (Figs. 9C and 9E). Phallic apparatus bilaterally symmetrical, curved downward, cylindrical, elongate, membranous apically (Figs. 9F–9H); apex elongate, in caudal view, phallobase tubular, endotheca cylindrical (Fig. 9G). Phallic spine and phallotremal sclerite absent (Figs. 9F–9H).

**Distribution.** Costa Rica (Guanacaste, Limón).

**Material examined. Holotype (male): COSTA RICA, Guanacaste,** Parque Nacional Guanacaste, ca. 0.7 km N Est. Maritza, 10.96°N, 85.50°W, 550 m, 31.viii.1990, Huisman and Quesada (UMSP); **Paratype: COSTA RICA, Limón**, Parque Nacional Braulio Carrillo, Quebrada González, 10.160°N, 83.939°W, 480 m, 12–14.v.1990, Holzenthal and Blahnik—one male (UMSP).

**Etymology.** From Latin *hasta* = spear, *pullus* = dark-colored, blackish. This is a reference to the sclerotized tips of the dorsolateral processes and tergum X.

*Oecetis plenuspinosa* Quinteiro & Holzenthal, n. sp. *urn:lsid:zoobank.org:act:9C9C4C21-B3F3-454C-B5B3-A1D9709ABBCF*

**Diagnosis.** This new species can be placed close to those of the *testacea*-group as defined by (*Malicky, 2005*), due to the presence of reticulate modifications on abdominal segments V-VIII. *Oecetis plenuspinosa* can be distinguished from the other species of *Oecetis* by the shape of the inferior appendage, which lacks dorsal and ventral lobes, by the ventral lobe of tergum X, with two lateral, posterior pointing, digitate projections, and by the distinctly clavate dorsal lobe of tergum X. Additionally, the phallic apparatus has 10 short spines, distributed symmetrically in two groups of five.

In the Neotropical region the only similar species, described to date, is *O. iara Henriques-Oliveira, Dumas & Nessimian, 2014*. *Oecetis plenuspinosa*, n. sp. differs from *Oecetis iara* due to its dorsal lobe of tergum X, with clavate apex, while *Oecetis iara* has the same structure broad basally, tapering toward an acute apex. Also, the new species has two digitate processes on the ventral margin of the ventral lobe of tergum X and a truncate apex on the inferior appendage, while *Oecetis iara* does not have these processes on tergum X and has the inferior appendage with the apex digitate.

**Description. Male:** forewing length 5 mm ($n = 1$).

**Head.** Color pale yellow (pinned specimen). Antennae three times length of forewing; scape stout, elongate; pedicel enlarged in width, subequal to other flagellomeres in length, shorter than scape. Maxillary palps yellow, 5-segmented, setose. Labial palps pale yellow, 3-segmented.

**Thorax.** Pterothorax yellowish brown; forewing yellow; dark bands over cord absent; dark spots absent on wing; forks I and V rooted; sectoral crossvein (*s*) not aligned with *r-m*. Hind wing with forks I and V present. Legs yellowish brown, mid leg with longitudinal row of spines on tibia and tarsal segments. Tibial spur formula 1,2,2.

**Abdomen.** Segments V, VI, VII and VIII with honeycomb texture on terga (Figs. 10A–10C), segment VIII with honeycomb cells smaller than others (Fig. 10C); segment IX annular short (Figs. 10D and 10E). Preanal appendage digitate, bearing apical setae (Figs. 10D and 10E). Tergum X, in lateral view, divided into dorsal and ventral lobes; dorsal lobe elongate, cylindrical, apex distinctly clavate (Figs. 10D and 10E); ventral lobe divided into two digitate sclerotized processes, slender, curved slightly upward, rounded tip (Figs. 10D and 10E). Inferior appendage 1-segmented, setose (Figs. 10D and 10F); dorsal lobe absent (Figs. 10D and 10F); ventral lobe rounded (Figs. 10D and 10F); distal lobe narrow, cylindrical, distal half enlarged, apex truncate; short and stout spine-like setae absent (Figs. 10D and 10F). Phallic apparatus bilaterally symmetrical, curved downward, cylindrical, distal half enlarged, apex truncate, pair of short processes dorsally, both slender, acuminate (Fig. 10G), ten phallic spines present, small, sickle shaped, simetrically distributed in two groups of five (Fig. 10G, one side represented); in caudal

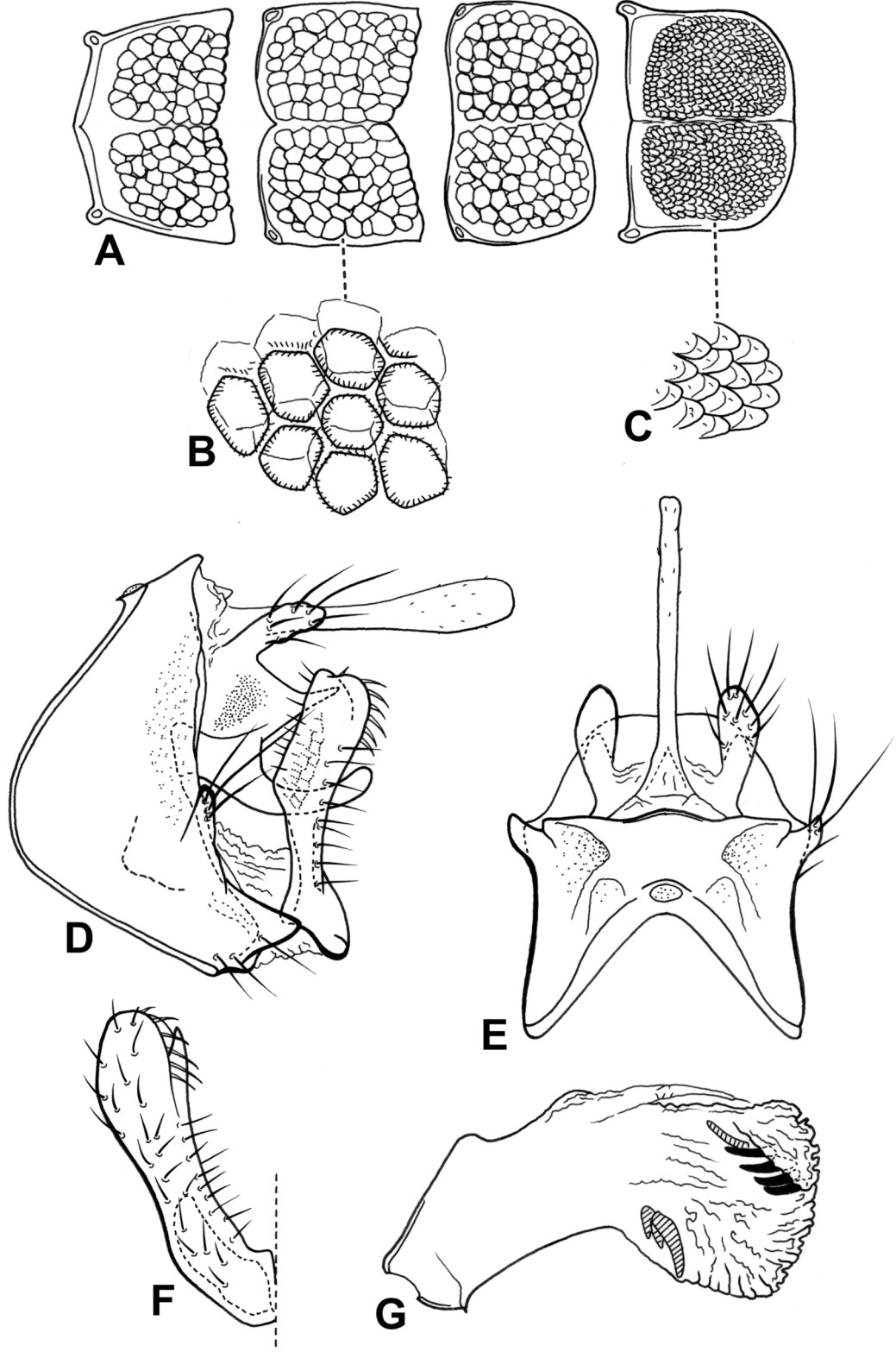

**Figure 10 Male genitalia of *Oecetis plenuspinosa* n. sp.** *Oecetis plenuspinosa* n. sp., Quinteiro & Holzenthal, male holotype. (A) abdominal terga V, VI, VII, VIII, dorsal view. (B) abdominal tergum VI texture, dorsal view. (C) abdominal tergum VIII texture, dorsal view. (D) genitalia, lateral view. (E) genitalia, dorsal view. (F) inferior appendage, ventral view. (G) phallic apparatus, lateral view.

view, with apex short, cylindrical, endotheca enlarged, bilobate laterally. Phallotremal sclerite present, horseshoe-shaped.

**Distribution.** Costa Rica (Limón).

**Material examined. Holotype (male): COSTA RICA, Limón,** E.A.R.T.H., forest reserve arroyo, 7.5 km (air) NW Pocora, 10°13′48″N, 083°33′36″W, 10, 4–5.ii.1992, Holzenthal, R.W., Muñoz, F., Kjer, K.M. (UMSP).

**Etymology.** From Latin *plenus* = full, plenty, *spinosus* = thorny. This is a reference to the many small spines present in the phallic apparatus.

*Oecetis machaera* Quinteiro & Holzenthal, n. sp. *urn:lsid:zoobank.org:act:ED8452F1-64C2-4432-8B55-CCCAD8E58DC1*

**Diagnosis.** This species is distinguished from the others in the *falicia* group by its bilobate inferior appendage, with its ventral lobe elongate, cylindrical, and apically acute.

*Oecetis machaera*, n. sp. is similar to *O. prolongata* Flint, 1981 due to the short, slightly ventrally bent, short dorsolateral process on segment IX. However, *Oecetis prolongata* has the ventral lobe of the inferior appendage absent. *Oecetis machaera*, n. sp. has the ventral lobe distinctly projected, cylindrical, and with an acute apex. Additionally, the phallic apparatus of *Oecetis prolongata* is very long and strongly bent ventrally, while the phallic apparatus of the new species is short and almost straight.

**Description. Male:** forewing length 4.8 mm ($n = 1$).

**Head.** Color yellowish brown (pinned specimen). Antennae 3.5 times length of forewing; scape stout, elongate; pedicel enlarged in width, subequal to other flagellomeres in length, shorter than scape. Maxillary palps yellowish brown, 5-segmented, setose.

**Thorax.** Pterothorax yellowish brown; forewing yellowish brown; dark bands over cord absent; dark spots absent; forks I and V rooted; sectoral crossvein (*s*) not aligned with *r-m*. Hind wing with forks I and V present. Legs yellowish brown, mid leg with longitudinal row of spines on tibia and tarsal segments. Tibial spur formula 0,2,2.

**Abdomen.** Segment IX annular, short, bearing pair of dorsolateral processes, each thick, straight, apex slightly bent ventrad, flattened on basis, tapering posteriorly, shorter than phallic apparatus (Fig. 11A). Preanal appendage long, digitate, bearing apical setae (Figs. 11A and 11B). Tergum X, in lateral view, divided into dorsal and ventral lobes (Fig. 11A); dorsal lobe modified into single cylindrical structure, apex digitate, shorter than ventral lobe, with short apical setae (Figs. 11A and 11B); ventral lobe divided medially by V-shape incision, broad basally, acute apically in dorsal view (Figs. 11A and 11B). Inferior appendage 1-segmented, setose (Figs. 11A and 11C); dorsal lobe absent (Fig. 11A); ventral lobe elongate, slightly shorter than distal lobe, apex acute (Figs. 11A and 11C); distal lobe narrow, tapering posteriorly, apex rounded in lateral view (Figs. 11A and 11C); short, stout spine-like setae absent (Figs. 11A and 11C). Phallic apparatus bilaterally symmetrical, curved downward, cylindrical, elongate, membranous apically (Fig. 11D); in caudal view, apex elongate, endotheca slightly enlarge in width apically; one projection

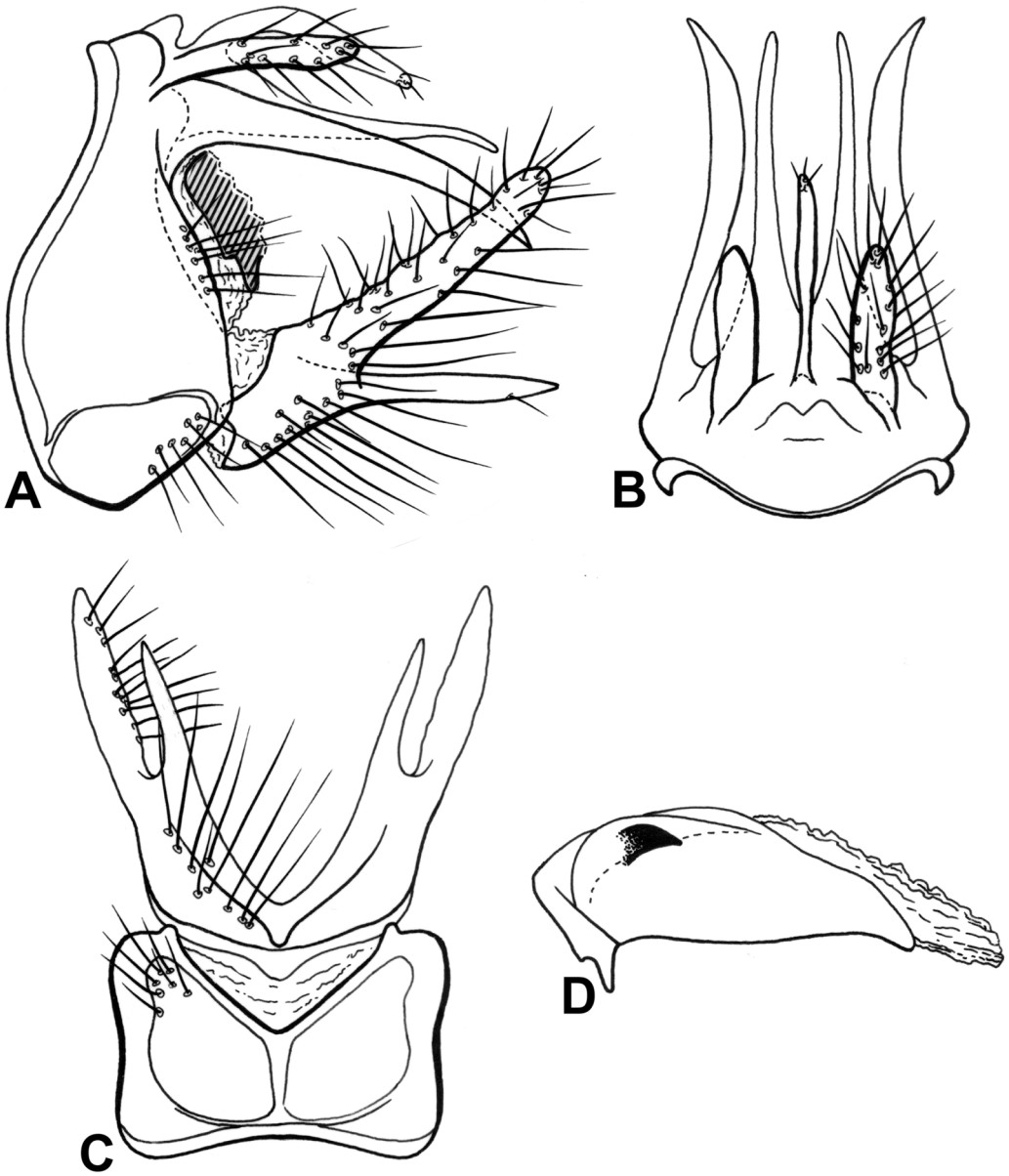

**Figure 11 Male genitalia of *Oecetis machaera* n. sp.** *Oecetis machaera* n. sp., Quinteiro & Holzenthal, male holotype. (A) genitalia, lateral view. (B) genitalia, dorsal view. (C) genitalia, ventral view. (D) phallic apparatus, lateral view.

on left side, acute, sclerotized (Fig. 11D). Phallic spine absent (Fig. 11D). Phallotremal sclerite horseshoe-shaped (Fig. 11D).

**Distribution.** Brazil (Amazonas).

**Material examined. Holotype (male): BRAZIL, Amazonas,** Am. 010, km 246, 20 km W Itacoatiara, 12–15.vii.1979, J. Arias et al. (NMNH, loan to UMSP).

**Etymology.** From Latin *machaera* = bent sword from ancient Greece, dirk, dagger. This specific epithet refers to the shape of the dorsolateral processes on segment IX in dorsal view, resembling the bent blade of some swords.

*Oecetis blahniki* Quinteiro & Holzenthal, n. sp. *urn:lsid:zoobank.org:act:9B6B3CF0-067A-4C51-B1CB-FD3CC15A08EF*

**Diagnosis.** This species can be distinguished from other *Oecetis* by a combination of characters. It has tergum X divided dorsoventrally, segment IX with two lateral rounded processes, projecting between the dorsal and ventral lobes of tergum X, the endotheca bilobed, and the inferior appendage with the ventral margin of the distal lobe angular, and a ventral lobe that is projecting and cylindrical, with a rounded apex.

This new species is similar to *O. gibbosa*, n. sp., *O. traini Rueda-Martín, Gibon & Molina, 2011*, and *O. rafaeli Flint, 1991* due to the presence of a distinct lateral process on segment IX and its short phallic apparatus. However, in *Oecetis traini* and *Oecetis rafaeli* the lateral processes are slender with acute apices, while in *Oecetis blahniki*, n. sp. and *Oecetis gibbosa*, n. sp. the apices are rounded, uniformly wide along their lengths, and project between the dorsal and ventral lobes of tergum X. The diagnostic difference between the two new species relies especially on the shape of the inferior appendage. *Oecetis blahniki*, n. sp. has the ventral margin of the distal lobe strongly angular and a projecting, cylindrical ventral lobe, while *Oecetis gibbosa*, n. sp. does not have a ventral lobe and its dorsal and distal lobes are terete and elongate. Additionally, *Oecetis blahniki*, n. sp has the endotheca bilobed, while in *Oecetis gibbosa*, n. sp. it is single lobed. This species does not have features to place it in any known species group.

**Description. Male:** forewing length 6.5 mm ($n = 1$).

**Head.** Color yellowish brown (pinned specimen). Scape stout, elongate; pedicel enlarged in width, subequal to other flagellomeres in length, shorter than scape. Maxillary palps yellowish brown, 5-segmented, setose. Labial palps yellow, 3-segmented.

**Thorax.** Pterothorax yellowish brown; forewing brown; dark bands over cord absent; dark spots absent; forks I and V rooted; sectoral crossvein (*s*) not aligned with *r-m*. Hind wing with forks I and V present. Legs yellowish brown, mid leg with longitudinal row of spines on tibia and tarsal segments. Tibial spur formula 0,2,2.

**Abdomen.** Segment IX annular, short, bearing pair of lateral processes, thick, cylindrical, slightly sinuous, tapering posteriorly, apex rounded, shorter than phallic apparatus (Figs. 12A and 12B). Preanal appendage short, rounded, apex somewhat pointing in dorsal view, bearing apical setae (Figs. 12A and 12B). Tergum X, in lateral view, divided in dorsal and ventral lobes (Figs. 12A and 12B); dorsal lobe modified into single cylindrical structure, apex acuminate, shorter than ventral lobe, with short apical setae (Figs. 12A and 12B); ventral lobe divided laterally by V-shaped incision, broad basally, digitate apically (Figs. 12A and 12B). Inferior appendage 1-segmented, setose (Figs. 12A and 12C); dorsal lobe absent (Fig. 12A); ventral lobe cylindrical, acuminate apex (Figs. 12A and 12C); distal lobe narrow, tapering posteriorly, apex acute, angular projection ventrally on mid region, apex acute (Figs. 12A and 12C); short, stout spine-like setae absent (Figs. 12A and 12C). Phallic apparatus bilaterally symmetrical, cylindrical, short, slightly curved ventrally (Fig. 12D); in caudal view, apex elongate. Endotheca longer

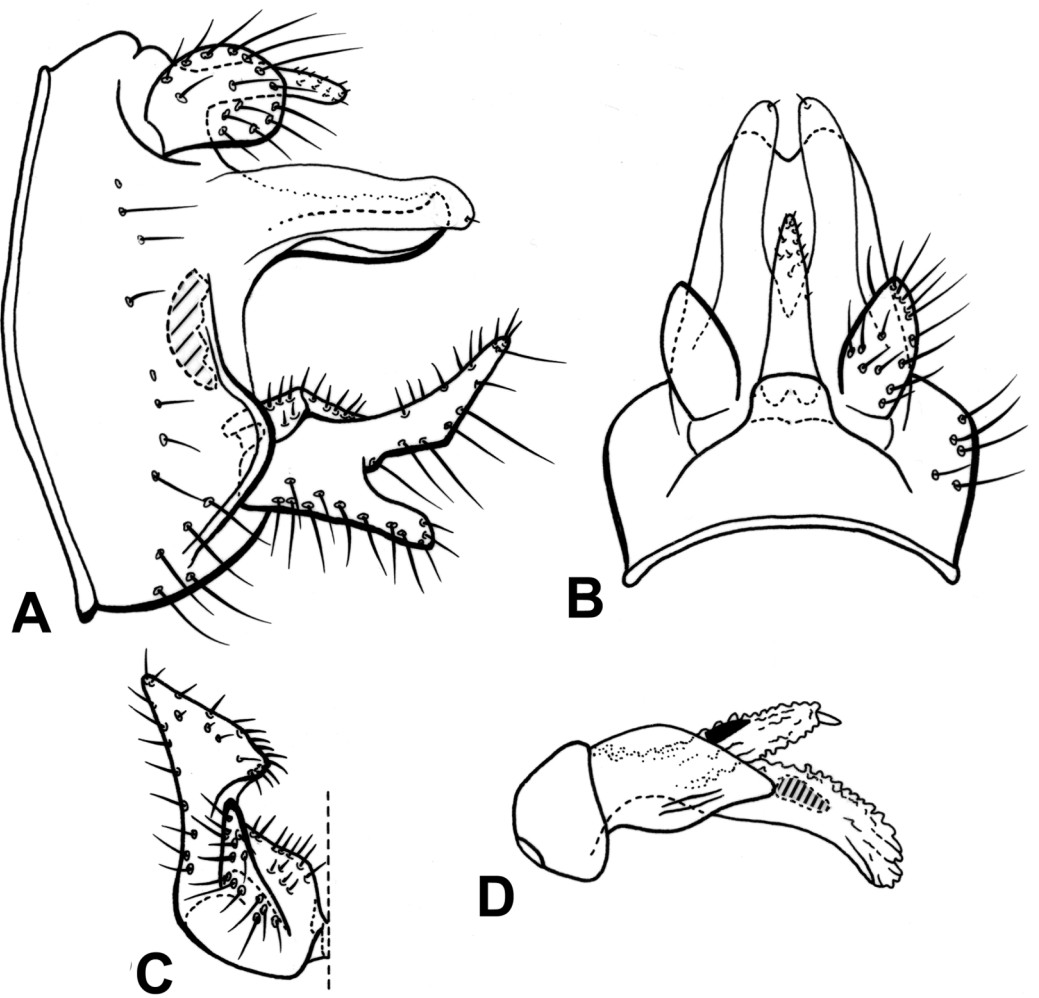

**Figure 12  Male genitalia of *Oecetis blahniki* n. sp.** *Oecetis blahniki* n. sp., Quinteiro & Holzenthal, male holotype. (A) genitalia, lateral view. (B) genitalia, dorsal view. (C) inferior appendage, ventral view. (D) phallic apparatus, lateral view.

than phallobase, bilobate (Fig. 12D). Two phallic spines present, straight. Phallotremal sclerite horseshoe-shaped.

**Distribution.** Brazil (Amazonas).

**Material examined. Holotype (male): BRAZIL, Amazonas,** Am. 010, km 246, 20 km W Itacoatiara, 12–15.vii.1979, J. Arias et al. (NMNH loan to UMSP).

**Etymology.** This specific epithet honors our colleague Roger J. Blahnik for his contributions to caddisfly taxonomy and systematics.

*Oecetis gibbosa* Quinteiro & Holzenthal, n. sp. *urn:lsid:zoobank.org:act:3118C6F6-3776-4624-B2E2-9615553CF62A*

**Diagnosis.** This species can be differentiated from the other *Oecetis* by its dorsoventrally divided tergum X, the presence of two lateral processes on segment IX with their apices

truncate and projecting between the lobes of tergum X, and by the inferior appendage with an elongate, cylindrical distal lobe and dorsal lobe that is slightly clavate apically and with a rounded, mesal lobe at midlength.

This new species is similar to *Oecetis traini Rueda-Martín, Gibon & Molina, 2011*, *O. rafaeli Flint, 1991*, and *O. blahniki*, n. sp. due to segment IX bearing a lateral process and also the short phallic apparatus. However, the lateral processes on segment IX in *Oecetis traini* and *Oecetis rafaeli* are slender, with acute apices, while the new species has them somewhat quadrate. *Oecetis gibbosa*, n. sp. differs from *Oecetis blahniki*, n. sp. in the lateral projections of segment IX, which have their apices truncate in *Oecetis gibbosa*, n. sp., while *Oecetis blahniki*, n. sp. has them rounded. Also, *Oecetis gibbosa*, n. sp. has the inferior appendage with elongate dorsal and distal lobes, and with an inner lobe on the dorsal lobe, while *Oecetis blahniki*, n. sp. does not have a developed dorsal lobe, and the distal lobe has a very angular ventral margin. Like some of the previous species presented here, this new species does not present any distinct characteristic that would allow us to place it in a species group.

**Description. Male:** forewing length 5 mm ($n = 1$).

**Head.** Color yellowish brown (pinned specimen). Scape stout, elongate; pedicel enlarged in width, subequal to other flagellomeres in length, shorter than scape. Maxillary palps yellowish brown, 5-segmented, setose. Labial palps yellow, 3-segmented.

**Thorax.** Pterothorax yellowish brown; forewing brown; dark bands over cord absent; dark spots absent; forks I and V sessile; sectoral crossvein (*s*) not aligned with *r-m*. Hind wing with forks I and V present. Legs yellowish brown. Tibial spur formula 0,2,2.

**Abdomen.** Segment IX annular, short (Figs. 13A and 13B). Preanal appendage long, digitate, bearing apical setae (Figs. 13A and 13B). Tergum X, in lateral view, divided into dorsal and ventral lobes (Figs. 13A and 13B); dorsal lobe modified into single cylindrical structure, apex digitate, nearly same length as ventral lobe, with short apical setae (Figs. 13A and 13B); ventral lobe undivided laterally, trapezoidal, smooth edges, apex truncate (Figs. 13A and 13B). Inferior appendage 1-segmented, broad basally, setose (Figs. 13A and 13C); dorsal lobe long, slender, apex clavate, with distinct rounded projection at mid region on mesal surface (Figs. 13A and 13A'), setae on inner surface of tip and mid region projection; ventral lobe absent (Figs. 13A and 13C); distal lobe narrow, tapering posteriorly, apex rounded, setose, forming 90° angle with dorsal lobe (Figs. 13A and 13C). Phallic apparatus bilaterally symmetrical, curved downward, cylindrical, elongate, membranous apically (Fig. 13D). One phallic spine present, slightly bent dorsally (Fig. 13D). Phallotremal sclerite absent (Fig. 13D).

**Distribution.** Brazil (Amazonas).

**Material examined. Holotype (male): BRAZIL, Amazonas,** Am. 010, km 229, 38 km W Itacoatiara, 29.i.1975, O.S. Flint Jr. (NMNH loan to UMSP).

**Etymology.** From Latin *gibbosus* = very humped. This species name is a reference to the projection observed at the mid region of the inferior appendage in caudal view.

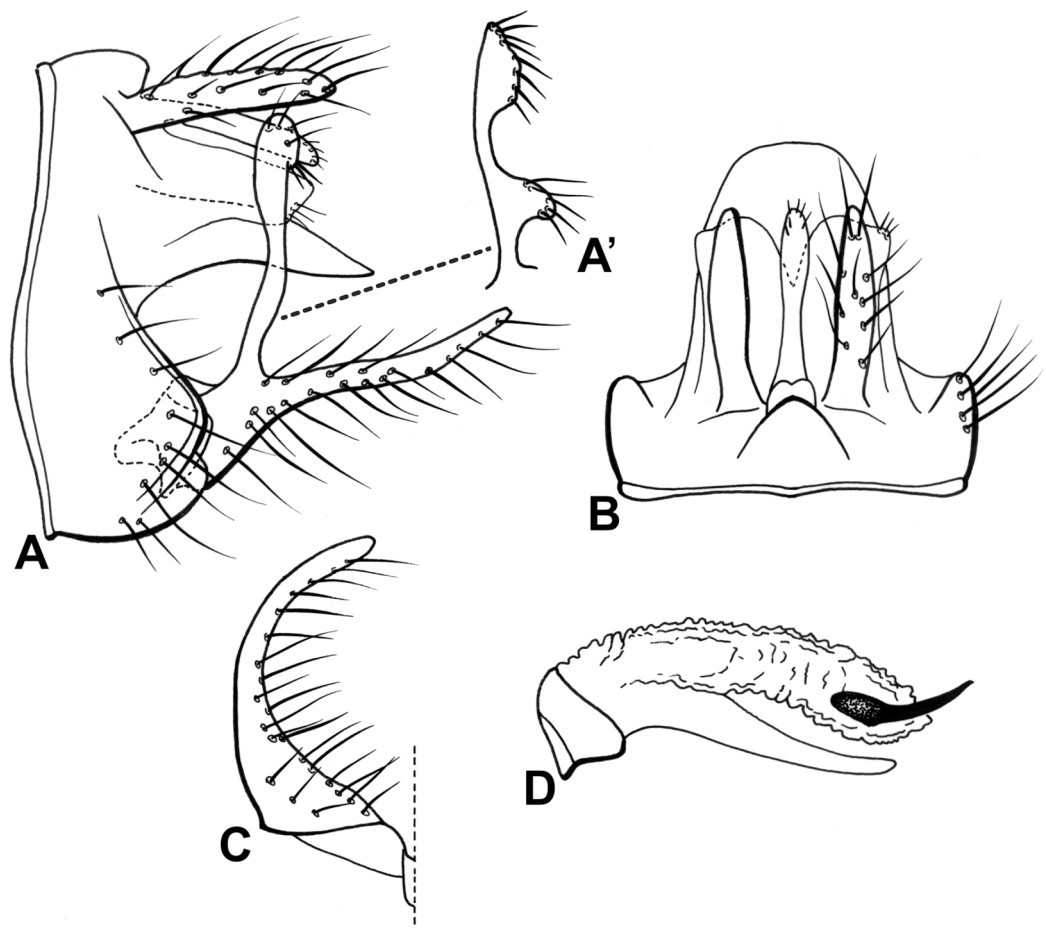

**Figure 13 Male genitalia of *Oecetis gibbosa* n. sp.** *Oecetis gibbosa* n. sp., Quinteiro & Holzenthal, male holotype. (A) genitalia, lateral view. (A') dorsal lobe of inferior appendage, caudal view. (B) genitalia, dorsal view. (C) inferior appendage, ventral view. (D) phallic apparatus, lateral view.

*Oecetis pertica* n. sp. Quinteiro & Holzenthal, n. sp. *urn:lsid:zoobank.org:act:3DA312EC-F046-4F36-8FD6-D61FF1D9E4AD*

**Diagnosis.** *Oecetis pertica* n. sp. can be distinguished from the other *Oecetis* by the long and cylindrical dorsal lobe of tergum X, together with the somewhat quadrate dorsal lobe of the inferior appendage with its apex truncate, and the presence of a protruding ventral lobe on the inferior appendage with a very angular margin and acute apex in lateral view. Also, the very conspicuous phallic spines are divided into two groups in different regions of the phallic apparatus. Finally, the quadrate lateral lobe on segment IX of this species is unique in *Oecetis*.

   This species has the inferior appendage similar to *O. doesburgi* (*Flint, 1974*), since both have a C-shaped incision between dorsal and distal lobes, and a dorsal lobe that is broad at its base, with its apex projecting distally. The new species has the dorsal lobe of inferior appendage with distinct truncate apex, whereas *Oecetis doesburgi* presents it rounded. Also, *Oecetis doesburgi* has the distal lobe enlarged apically, while *Oecetis pertica* n. sp. has
it narrow. The long dorsal lobe of tergum X and the mesally divided ventral lobe of tergum X of this new species are similar to *O. prolongata Flint, 1981*, but *O. pertica* n. sp. does not have the dorsolateral processes on segment IX that are diagnostic of the *falicia*-group and present in *Oecetis prolongata*. Finally, *O. rafaeli Flint, 1991*, and *O. blahniki* n. sp. also have a pair of lateral processes on segment IX, as well as *Oecetis pertica* n. sp., but neither of them has the lateral process quadrate, as in the new species. This new species does not have diagnostic characters that allow us to place it in a known species group.

**Description. Male:** forewing length 4.5 mm ($n = 1$).

**Head.** Color yellowish brown (pinned specimen). Scape stout, elongate; pedicel enlarged in width, subequal to other flagellomeres in length, shorter than scape. Maxillary palps yellowish brown, 5-segmented, setose. Labial palps yellow, 3-segmented.

**Thorax.** Pterothorax yellowish brown; forewing yellowish; small patches of dark setae present at junction of most veins, with patches of white setae adjacent to these; forks I and V sessile; sectoral crossvein (*s*) not aligned with *r-m*. Hind wing with forks I and V present. Legs yellowish brown. Tibial spur formula 0,2,2.

**Abdomen.** Segment IX uneven dorsoventrally, with anterior margin projecting midlaterally, tergum IX noticeably shorter than sternum IX, segment bearing pair of broad, slightly quadrate lateral processes from the posterior margin, projecting underneath ventral lobe of tergum X; acrotergite absent (Figs. 14A and 14B). Preanal appendage long, digitate, bearing apical setae (Figs. 14A and 14B). Tergum X, in lateral view, divided into dorsal and ventral lobes (Figs. 14A and 14B); dorsal lobe modified into single cylindrical structure, apex slightly clavate, nearly same length as ventral lobe, with short apical setae (Figs. 14A and 14B); ventral lobe divided medially by V-shape incision, broad basally, apex acute (Fig. 14B). Inferior appendage 1-segmented, broad basally, setose (Figs. 14A and 14C); dorsal lobe slightly quadrate, projecting distally, apex truncate (Fig. 14A); ventral lobe slightly protruding basally, keeled, apex acute and margin very angular in lateral view (Fig. 14A), and broadly rounded in ventral view (Fig. 14C); distal lobe broad, tapering distally, slightly bent dorsad, apex rounded, forming with dorsal lobe shallow C-shaped incision; short and stout spine-like setae absent (Figs. 14A and 14C). Phallic apparatus bilaterally symmetrical, curved ventrally, tubular, elongate, membranous apically, constricted at mid portion, enlarged distally (Fig. 14D). Endotheca with approximately 12 short, straight, phallic spines, 5–7 at mid region, 5–7 at apex (exact number difficult to discern, endothecal membranes not everted on specimens examined) (Fig. 14D). Phallotremal sclerite absent (Fig. 14D).

**Distribution.** Brazil (Amazonas).

**Material examined. Holotype (male): BRAZIL, Amazonas,** Am. 010, km 229, 38 km W Itacoatiara, 29.i.1975, O.S. Flint Jr. (NMNH).

**Etymology.** From Latin *pertica* = long pole. This species name refers to the long dorsal lobe of tergum X.

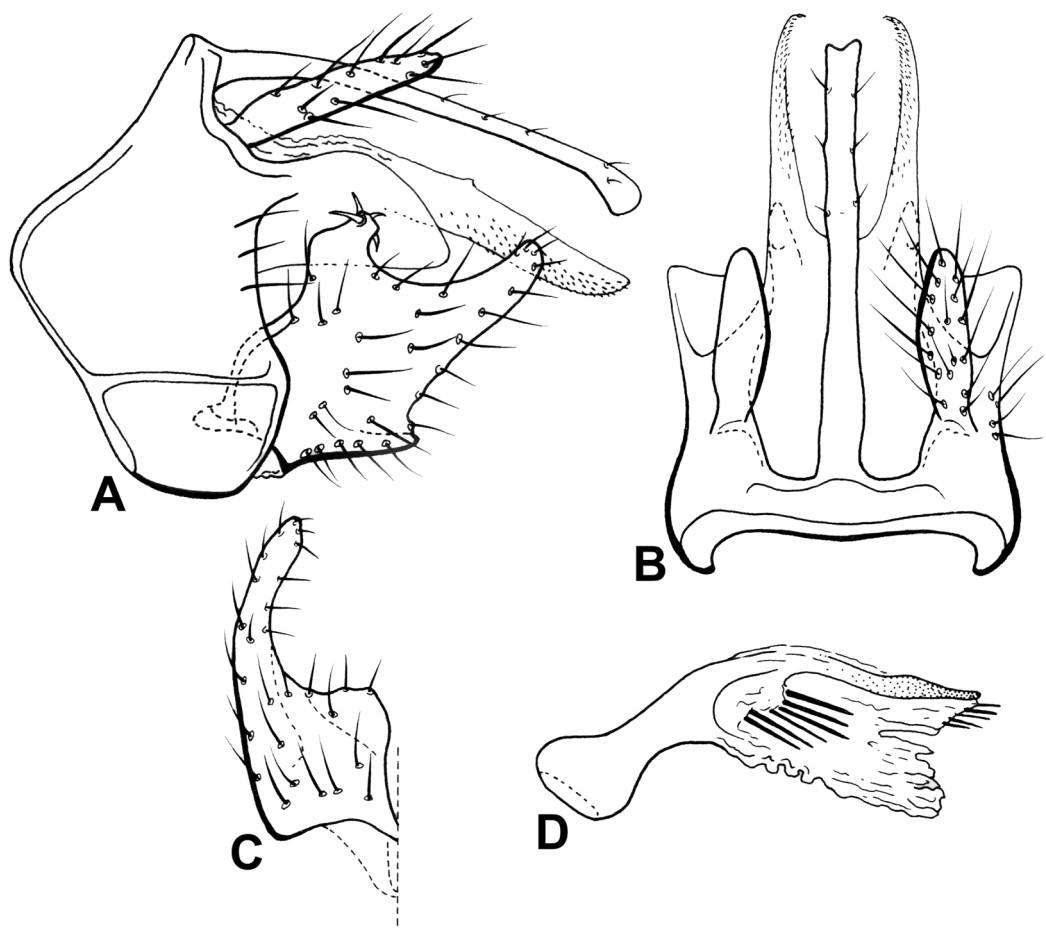

**Figure 14 Male genitalia of *Oecetis pertica* n. sp.** *Oecetis pertica* n. sp., Quinteiro & Holzenthal, male holotype. (A) genitalia, lateral view. (B) genitalia, dorsal view. (C) inferior appendage, ventral view. (D) phallic apparatus, lateral view.

*Oecetis licina* n. sp. Quinteiro & Holzenthal, n. sp. *urn:lsid:zoobank.org:act:9BFC14F7-FCD1-4FF3-AA38-F62B52439CCD*

**Diagnosis.** This species does not have characters that allow us to place it in a diagnosed species group. However, it can be distinguished from the other *Oecetis* by the enlarged dorsal lobe of tergum X, the triangular lateral process of segment IX, and the distinctly curved inferior appendage.

    *Oecetis licina* n. sp. is similar to *Oecetis gibbosa* n. sp., since they both share a segment IX lateral process protruding underneath tergum X, as well as a cylindrical dorsal lobe of tergum X and a phallic apparatus that is cylindrical, slightly curved ventrally and bearing one phallic spine. *Oecetis licina* n. sp. is also similar to *Oecetis blahniki* n. sp. based on the broad, mesally curved distal lobe of the inferior appendage and the rounded preanal appendage. However, *Oecetis licina* n. sp. and *Oecetis gibbosa* n. sp. differ greatly in the inferior appendage shape. *Oecetis gibbosa* n. sp. has the dorsal lobe of inferior appendage terete, with apical and mesal inner projections. *Oecetis licina* n. sp. has the dorsal lobe of inferior appendage discreetly projected and rounded. The distal lobe of the inferior

appendage in *Oecetis licina* n. sp. is conspicuously enlarged compared to the terete distal lobe of inferior appendage in *Oecetis gibbosa* n. sp. Also, the phallic apparatus in *Oecetis licina* n. sp. is disproportionally large compared to the remainder of the genitalia. Compared to *Oecetis blahniki*, n. sp., *Oecetis licina* n. sp. has the ventral lobe of inferior appendage absent, whereas *Oecetis blahniki* n. sp. has it cylindrical, with acuminate apex. Also, the lateral processes on segment IX of *Oecetis licina* n. sp. are broad at base and triangular, with acute apex, while *Oecetis blahnik* n. sp. has them cylindrical throughout, with rounded apex.

**Description. Male:** forewing length 5.5 mm (*n* = 1).

**Head.** Color yellowish brown (pinned specimen). Scape stout, elongate; pedicel enlarged in width, subequal to other flagellomeres in length, shorter than scape. Maxillary palps yellowish brown, 5-segmented, setose. Labial palps yellow, 3-segmented.

**Thorax.** Pterothorax yellowish brown; forewing light brown; faint band over cord; dark spots apically; forks I and V sessile; sectoral crossvein (*s*) not aligned with *r-m*. Hind wing with forks I and V present. Legs yellowish brown. Tibial spur formula 0,2,2.

**Abdomen.** Segment IX annular, with short, triangular lateral processes present, broad basally, acute apically, projecting underneath dorsal lobe of tergum X; acrotergite present as two structures dorsolaterally (Figs. 15A and 15B). Preanal appendages short, rounded, bearing apical setae (Figs. 15A and 15B). Tergum X, in lateral view, divided into dorsal and ventral lobes; dorsal lobe modified into single structure, digitate, inflated preapically, subacute apically, same length as ventral lobe, with short setae (Figs. 15A and 15B); ventral lobe membranous, divided near apex by shallow, V-shaped incision, apex rounded (Fig. 15B). Inferior appendage 1-segmented, broad basally, setose (Figs. 15A and 15C); dorsal lobe smoothly projected, rounded, discrete (Fig. 15A); ventral lobe absent; distal lobe, as viewed laterally broad basally, tapering distally, bent dorsally at mid region, keeled ventrally, ventral margin distinctly angular, apex mesally curved and rounded (Figs. 15A and 15C); short and stout spine-like setae present on dorsal lobe and apical inner portion (Figs. 15A and 15C). Phallic apparatus bilaterally symmetrical, elongate, cylindrical, curved ventrally (Fig. 15D). One phallic spine present, straight (Fig. 15D). Phallotremal sclerite absent (Fig. 15D).

**Distribution.** Guyana

**Material examined. Holotype (male): GUYANA,** Essequebo [sic] R., Br. Guiana, July 1921, A. Busck coll [verbatim, no additional information given] (NMNH).

**Etymology.** From Latin *licinus* = bent or turned upward. This species name refers to the distal lobe of inferior appendage bent dorsally.

## Concluding remarks

This study raises the number of *Oecetis* in the Neotropics from 55 to 69 species. The new species distributions are summarized in Fig. 16. It is noticeable that some of them are only known by their holotype specimen since it was the only material available so far. Although this is not the ideal situation, we choose to describe these new species instead

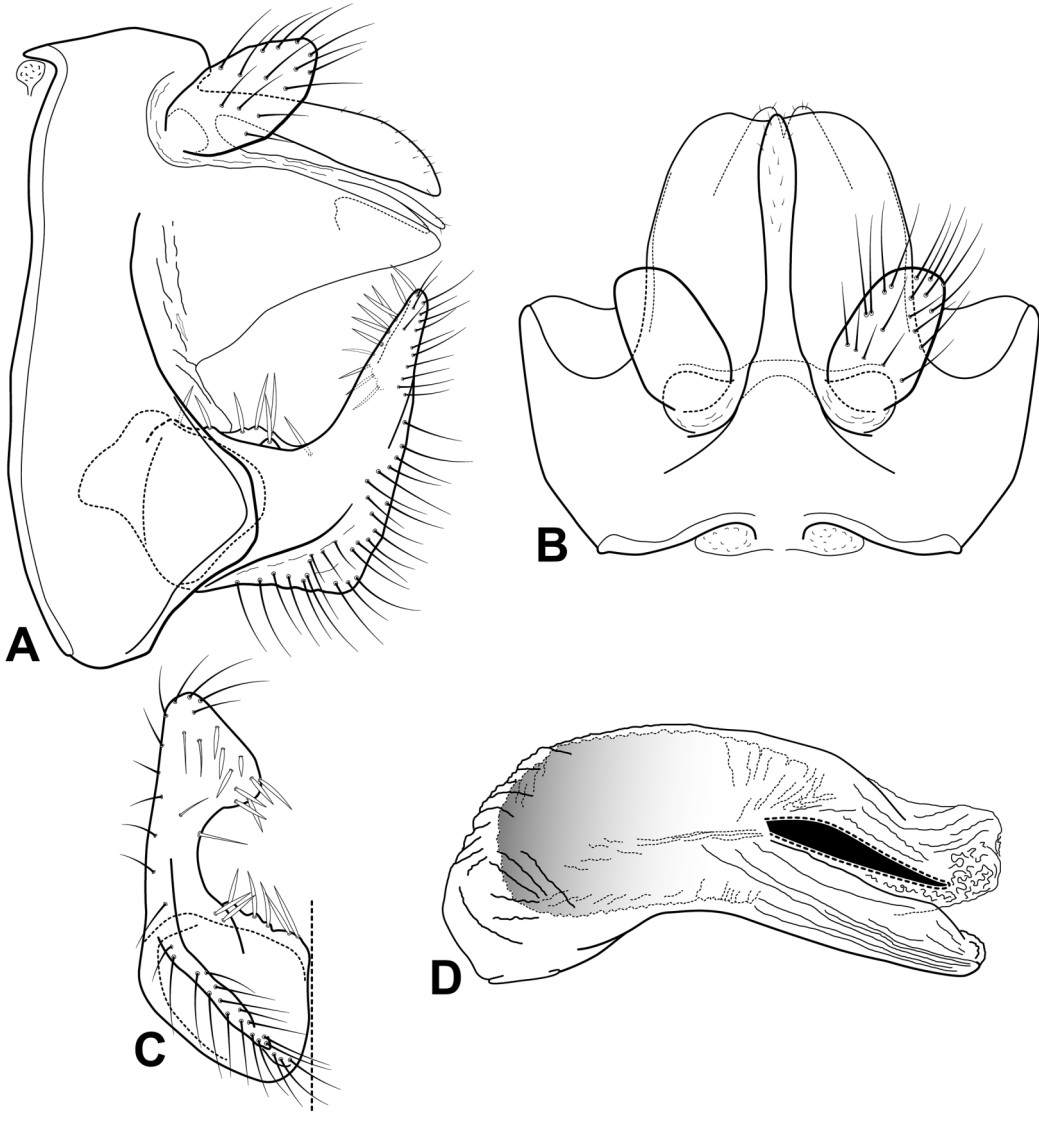

**Figure 15 Male genitalia of *Oecetis licina* n. sp.** *Oecetis licina* n. sp., Quinteiro & Holzenthal, male holotype. (A) genitalia, lateral view. (B) genitalia, dorsal view. (C) inferior appendage, ventral view. (D) phallic apparatus, lateral view.

of letting them sit in museum collections for up to 20 years, as it can be seen in the case of *Oecetis pertica*, n. sp. We hope in the future, new information about their behavior or morphological variation can be provided as additional specimens are observed and collected.

Even with our contribution on the Neotropical diversity of *Oecetis*, many other questions remain unanswered. In this study, eight new species do not present diagnostic characters that allow us to place them in the already proposed taxonomic groups. This may be an indication that much of the diversity of the genus is still to be discovered, especially in Amazonia where many unexplored areas may harbor new species.

Another issue is the absence of a phylogenetic hypothesis of *Oecetis* species. It has been suggested that the *avara*- and *punctata*-groups are closely related (*Blahnik & Holzenthal,*

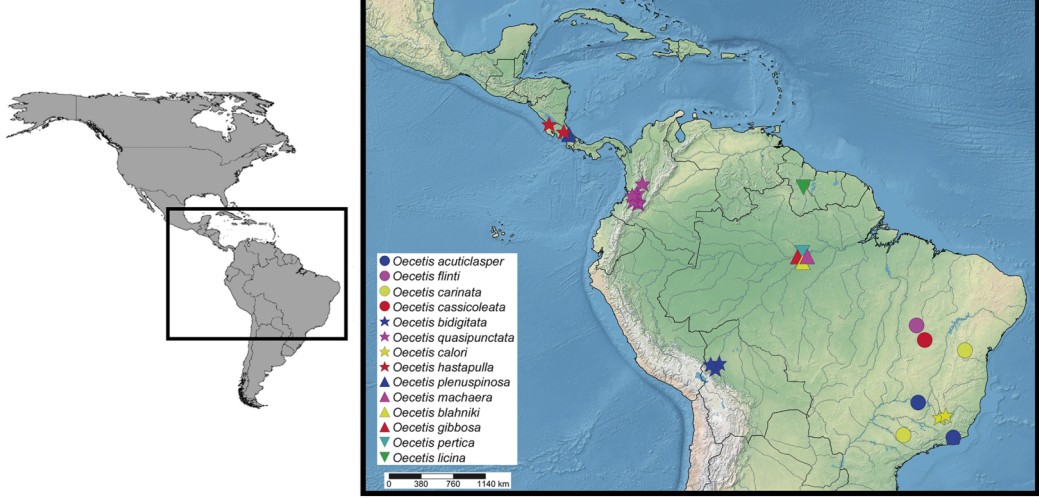

**Figure 16 Distribution of the 14 new species of *Oecetis* based on the specimens presented on material examined sections.**

*2014*). Other than this, there is no further phylogenetic information on *Oecetis* species or groups of species. Since the phylogenetic relationships among species remain unclear, it is difficult to determine the placement of certain species to species group, for example. In this way, a phylogenetic study would be of much value to identify the character diversity present in *Oecetis* and how these characters are related to each other. Also, a phylogenetic hypothesis should properly evaluate the delimitation of the already proposed taxonomic groups (e.g., *avara-*, *inconspicua*-groups). Considering that some of the new species described here do not fit in any species group diagnosis, perhaps a re-delimitation of those groups is necessary. Since the genus currently contains more than 500 species divided in no more than a dozen species groups, a phylogenetic hypothesis becomes essential to a stable taxonomy with well circumscribed species groups.

## ACKNOWLEDGEMENTS

We are very grateful to Dr. Roger Blahnik, Dr. Jane Hughes, and an anonymous reviewer for very insightful suggestions that improved the manuscript. We are grateful to Dr. Oliver S. Flint and Dr. Adolfo R. Calor for the generous loan of specimen loans. We appreciate the helpful suggestions on this manuscript of Adolfo R. Calor, Daniela M. Takiya, and Eduardo A. B. Almeida.

### Funding

This work was supported by the National Counsel of Technological and Scientific Development (CNPq; process 142211/2012-5 to FBQ), Coordination for the Improvement of Higher Education Personnel (CAPES; process BEX 14209/13-6 to FBQ) and São Paulo Research Foundation (FAPESP; process 2011/09477-9 to Eduardo

Almeida). Support was also provided from the University of Minnesota, Agricultural Experiment Station projects 017-17 and 017-29. The funders had no role in study design, data collection and analysis, decision to publish, or preparation of the manuscript.

### Grant Disclosures

The following grant information was disclosed by the authors:
National Counsel of Technological and Scientific Development: 142211/2012-5.
Coordination for the Improvement of Higher Education Personnel: 14209/13-6.
São Paulo Research Foundation: 2011/09477-9.
University of Minnesota, Agricultural Experiment Station: 017-17 and 017-29.

### Competing Interests

The authors declare that they have no competing interests.

### Author Contributions

- Fabio B. Quinteiro conceived and designed the experiments, performed the experiments, analyzed the data, contributed reagents/materials/analysis tools, wrote the paper, prepared figures and/or tables, reviewed drafts of the paper.
- Ralph W. Holzenthal conceived and designed the experiments, performed the experiments, analyzed the data, contributed reagents/materials/analysis tools, wrote the paper, prepared figures and/or tables, reviewed drafts of the paper.

### Data Availability

  The specimens are deposited in museums as vouchers.

### New Species Registration

The following information was supplied regarding the registration of a newly described species:
Publication LSID: urn:lsid:zoobank.org:pub:ED02CA58-B074-45A6-AAC7-48FB48B97BA8
New taxa LSIDs:
*Oecetis acuticlasper* Quinteiro & Holzenthal, n. sp. urn:lsid:zoobank.org:act:046E520D-07ED-4892-BBDE-BE0654C5BE95
*Oecetis flinti* Quinteiro & Holzenthal, n. sp. urn:lsid:zoobank.org:act:E760A8EB-7908-427C-AC19-D11291E15FE8
*Oecetis carinata* Quinteiro & Holzenthal, n. sp. urn:lsid:zoobank.org:act:404BC99D-A18C-4322-892F-E824DA3B66CF
*Oecetis cassicoleata* Quinteiro & Holzenthal, n. sp. urn:lsid:zoobank.org:act:140E12D6-B022-4128-9AB9-BF148264378F
*Oecetis bidigitata* Quinteiro & Holzenthal, n. sp. urn:lsid:zoobank.org:act:7A089FD8-F3F1-4339-898F-98DE402E3C81
*Oecetis quasipunctata* Quinteiro & Holzenthal, n. sp. urn:lsid:zoobank.org:act:B7E84B92-234B-46F1-BDF0-9D74D1CA9AB9

*Oecetis calori* Quinteiro & Holzenthal, n. sp. urn:lsid:zoobank.org:act:58B08D3F-32E2-4408-9D62-24A46FAB2B5F

*Oecetis hastapulla* Quinteiro & Holzenthal, n. sp. urn:lsid:zoobank.org:act:B43E4DCE-3579-4BDD-B803-B32151AB0327

*Oecetis plenuspinosa* Quinteiro & Holzenthal, n. sp. urn:lsid:zoobank.org:act:9C9C4C21-B3F3-454C-B5B3-A1D9709ABBCF

*Oecetis machaera* Quinteiro & Holzenthal, n. sp. urn:lsid:zoobank.org:act:ED8452F1-64C2-4432-8B55-CCCAD8E58DC1

*Oecetis blahniki* Quinteiro & Holzenthal, n. sp. urn:lsid:zoobank.org:act:9B6B3CF0-067A-4C51-B1CB-FD3CC15A08EF

*Oecetis gibbosa* Quinteiro & Holzenthal, n. sp. urn:lsid:zoobank.org:act:3118C6F6-3776-4624-B2E2-9615553CF62A

*Oecetis pertica* n. sp. Quinteiro & Holzenthal, n. sp. urn:lsid:zoobank.org:act:3DA312EC-F046-4F36-8FD6-D61FF1D9E4AD

*Oecetis licina* n. sp. Quinteiro & Holzenthal, n. sp. urn:lsid:zoobank.org:act:9BFC14F7-FCD1-4FF3-AA38-F62B52439CCD

## Supplemental Information

Supplemental information for this article can be found online at http://dx.doi.org/10.7717/peerj.3753#supplemental-information.

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
