# Peer review of "Fourteen new species of Oecetis McLachlan, 1877 (Trichoptera: Leptoceridae) from the Neotropical region"

_PeerJ, doi:10.7717/peerj.3753_

## Round 0.1 · original submission · Major Revisions

Two reviewers have assessed this manuscript and both see that it is potentially of interest to readers of the journal.

However, there are a number of issues that would need to be addressed.

First, it is not clear from the introduction why this work is important and how it fits into previous work.
Second, the number of specimens examined of each species is not clear. Moreover, it seems that some species are described from a single specimen. This seems risky, as there is no data on how much variation there might be within species. There is no molecular data to back up any of the species designations, so there needs to be more justification of these descriptions based just on morphology.
As noted by reviewer 2, PeerJ might not be the most appropriate journal for a paper merely describing new species, as there is currently little justification of the need for the work, and no additional evidence in the form of DNA data or behaviour for example for the species designations.
Third, the paper needs a discussion section. Reviewer 1 has made some suggestions as to what might go into this.

There are a number of other comments made by reviewers which shoudl be addressed in any revision.

Both reviewers have provided very constructive comments on the paper, both in the comments sections, and, in the case of reviewer 2, on the ms itself. This should assist authors in improving the paper.

Reviewer 1 ·

Basic reporting

Basic Reporting:

The manuscript describes 14 new species of Oecetis and provides basic alpha taxonomy for the Neotropics. As such it is well written and descriptions of the adult males are adequate, although there appears to be few specimens on which the descriptions are based. The authors do not state how many animals were used per species. The English expression is sound, but there may need to be consideration of the order of some sentences; see in general comments.

The Introduction and Background are suitable for the publication and show how this work is relevant to the biodiversity of the area. It also highlights the relationships between Oecetis spp and sub-groups within the genus. The literature is relevant and has been well reviewed.

Figures are relevant and in most cases consistently presented enabling the reader to compare and contrast the characters directly.

Experimental design

As a publication of alpha taxonomy, which is essential in areas in which the insect fauna that are poorly known, it is unclear why the authors selected PeerJ for this manuscript. It is original primary research but lacks any phylogenetic considerations or even molecular supporting evidence of the species validity. It is very basic work and there are a number of other journals that would seem to be more appropriate for this work.

The research question is poorly defined and considering the considerable work by the second author in determining the caddisfly diversity in the Neotropical Region over many years this work needs to be put in this context. There is a clear need for the collections and descriptions of new species, but all that is provided in Line 96 was “In an attempt to discover the diversity of the Neotropical Region ….”. Prof. Holzenthal’s work is far more than this and deserves more defined research aims and objectives.

The investigation and methods were literally place Malaise traps and light traps and collect specimens. Technically very basic, but essential to collect material. These basic methods are adequate to enable repeatability.

A map of the sampling locations would be an improvement for readers who do not know the areas sampled.

Validity of the findings

14 new species are described, but it is unclear how many specimens from each species were collected or used for the descriptions. No comment is passed on variability for any species. New species based on a single specimen can lead to taxonomic confusion if the species has a high degree of variability of character expression. Some comments on potential issue with variability should be made where only a single specimen has been described.

Within each species the comparisons with described species is well done and on the basis of the descriptions differences are well presented. Variation again is not mentioned.

There is no discussion presented even though there are some interesting questions from this research. In the Introduction the six species groups within the genus Oecetis are described, but 8 of the 14 new species do not have characteristics allowing them to be placed in any of these groups. Given this there needs to be a discussion on the value of these groups; are there relevant new groups or are the groups of limited taxonomic value? Are the groups of limited value in the Neotropical region? Molecular analyses would be useful in the determination of their phylogenetic relationships and hence the value of the groups. Speculation along the phylogenetic relationships could have been included in the discussion.

How does this research improve knowledge of Neotropical Caddisflies? Is it a significant contribution?

Additional comments

Lines 92-95: Suggest re-wording to “There are species already deposited and labeled in museums waiting to be described. This study advances the knowledge of the Neotropical caddisfly diversity by describing fourteen new species of Oecetis based on morphological characteristics of the adult male.”

Line 102: delete “the specimens had their” and after genitalia insert “were”

Lines 104-5: reword “vials in approximately 50uL of glycerin.”

Line 107: delete “rendered”.

Line 135 and in all descriptions of the 8 species which do not conform to one of the “species groups” the statement “This species does not have features to place it any known species group.” Is not a diagnosis. It needs to be at the end of the diagnosis or at the end of the comparison section. Such as at the end of Line 147.

Line 148: Description. How many specimens used in this description? Is it of the holotype or just a male? This comment is relevant for all descriptions.

Line 179: as above for Line 135.
Line 182: Delete “However,”

Lines 221-222: As above.

Line 221: Delete “However,”

Line 231: replace “is” with “in”

Line 260: as above.

Line 261: Delete “However,” and change “diagnosed” to “distinguished”

Line 300-301: Delete the sentence starting “the three of them…”.

Line 301: Change “differently than” to “differs from”

Line 302: Delete “had a” and replace with “by having a”

Line 340: Change “diagnosed” to “distinguished”

Lines 522-530: Good comment and discussion, but needs to be part of a discussion after the descriptions.

Line 622: change “diagnosed” to “distinguished” or “differentiated”

Line 666: change “diagnosed” to “distinguished”

Line 674: Delete “Differently of O. doesburgi”

Line 675: Change sentence to “… distinct truncated apex and in O. doesburgi it is rounded.”

Line 679: After IX insert “which is”

Line 716-717: As above for Line 135. And delete “However,”

Line 726 “distal lobe of the inferior appendage…”

Line 727: “terete distal lobe of the inferior…”

Acknowledgements: How was this collecting trip funded? Was there a funding body? If so needs to be acknowledged.

Line 778: Chen 1993 is an unpublished thesis so strictly speaking the date should be unpublished and 1993 after the thesis details.

·

Basic reporting

Some reediting of the manuscript, with respect to grammar and clarity is warranted. Suggestions and comments are included in a marked up version of the manuscript and in the general comments to the authors below.

Experimental design

No comment

Validity of the findings

The species all appear to be new species.

Additional comments

Review: Fourteen new species of Oecetis McLachlan, 1877 (Trichoptera: Leptoceridae) from the Neotropical region
Authors: Fabio B. Quinteiro1 and Ralph W. Holzenthal
I am recommending that the paper be accepted, with a moderate revision. In describing 14 new species from the Neotropics, the paper makes an important contribution to the taxonomy of the genus. The illustrations are somewhat variable, but probably adequate to identify the new species. However, the accompanying diagnoses and descriptions need some work, both with respect to grammar and technical accuracy. I have made some editorial suggestions and comments in the .docx version of the manuscript, but a more thorough reediting is warranted. I especially recommend that the junior author review the grammar before resubmission. I do have several additional comments, with respect to content. A number of the species are assigned to species groups, but there is no clear indication, either by list or reference to previous literature, what the membership of the groups is. The defining characters of the groups are only very generally discussed for some groups. It is possible that the groups are primarily erected to consider Neotropical species and that the membership for species outside the Neotropics is not definitively known. If so, this should be clearly stated. However, the Neotropical membership of the groups should still be clearly indicated. There is no value in assigning a species to a group, if its membership is obscure.
One of the species described is compared to O. punctata. I know of no published illustration of this species. It is very difficult to diagnose a species by verbal descriptions alone and I would recommend that the manuscript include an illustration of O. punctata, in addition to the new species.
In reading the descriptions of the species, I found the description of the inferior appendage to be most difficult to follow. The inferior appendage is especially diagnostic, and extremely variable in form. The descriptions are written so as to conform to a ground plan in which the appendage has a dorsal, ventral, and distal appendage. However, it was not clear (to me) that all of the species actually follow such a ground plan. Even if they do, it is not always easy to relate a given species to this basic ground plan and thus the descriptions are often difficult to follow. I would recommend that the appendage be described in more general terms. Also, the description of tergum X is also sometimes difficult to follow. My interpretation is that the digitate and setose dorsal lobe corresponds to tergum X, as it occurs in other Trichoptera, and that the lateral lobes are (more or less) sclerotized extensions of the periphallic membrane. The absence of setae on these lobes would be consistent with this interpretation. However, I have noted that these are often referred to as lateral lobes of tergum X in other descriptions of Oecetis, and so I can’t argue with the use of the terminology followed in this paper. Assuming that these lobes do straddle the phallus, it was not always clear, either from the illustrations or descriptions, whether the cleft in the lobes was from the ventral margin, or whether the lobes may sometimes be continuous ventrally. I simply suggest that attention is paid to this issue in the restructuring of the descriptions.

---

## Round 0.2 · Minor Revisions

This paper is much improved following revisions.

a few minor issues have been identified by one of the reviewers and should be amended before the paper can be accepted.

Reviewer 1 ·

Basic reporting

No Comment

Experimental design

No Comment

Validity of the findings

No Comment

Additional comments

Fourteen new species of Oecetis McLachlan, 1877 (Trichoptera: Leptoceridae) from the Neotropical region

Authors: Fabio B. Quinteiro1 and Ralph W. Holzenthal2

The authors have addressed all my previous concerns. The manuscript is acceptable for publication with a couple of very minor changes.

Abstract: Background – reticulata group was deleted throughout the manuscript and replaced by testacea group, so this needs changing in the Background and Results and Discussion.

Chen is an unpublished thesis and should be cited as such, not 1933. This is on Lines 47, 64, 67 and 75.
Lines 167, 171, 172, 650 1nd 651 – Figs requires a period ie Figs.
Lines 604, 605, 607 Fig. Refers to multiple figures so should be Figs.
Line 772. Space needs to be inserted “discovered,especially” to “discovered, especially”

---

## Round 0.3 · accepted · Accept

This paper is now suitable for publication.